# Plasmodium myosin A drives parasite invasion by an atypical force generating mechanism

Julien Robert-Paganin[1], James P. Robblee[2], Daniel Auguin [3], Thomas C. A. Blake[4], Carol S. Bookwalter[2], Elena B. Krementsova[2], Dihia Moussaoui[1], Michael J. Previs[2], Guillaume Jousset[1], Jake Baum [4], Kathleen M. Trybus[2] & Anne Houdusse [1]

*Plasmodium* parasites are obligate intracellular protozoa and causative agents of malaria, responsible for half a million deaths each year. The lifecycle progression of the parasite is reliant on cell motility, a process driven by myosin A, an unconventional single-headed class XIV molecular motor. Here we demonstrate that myosin A from *Plasmodium falciparum* (PfMyoA) is critical for red blood cell invasion. Further, using a combination of X-ray crystallography, kinetics, and in vitro motility assays, we elucidate the non-canonical interactions that drive this motor's function. We show that PfMyoA motor properties are tuned by heavy chain phosphorylation (Ser19), with unphosphorylated PfMyoA exhibiting enhanced ensemble force generation at the expense of speed. Regulated phosphorylation may therefore optimize PfMyoA for enhanced force generation during parasite invasion or for fast motility during dissemination. The three PfMyoA crystallographic structures presented here provide a blueprint for discovery of specific inhibitors designed to prevent parasite infection.

[1] Structural Motility, UMR 144 CNRS/Curie Institute, 26 rue d'ulm, 75258 Paris cedex 05, France. [2] Department of Molecular Physiology and Biophysics, University of Vermont, Burlington, VT 05405, USA. [3] Laboratoire de Biologie des Ligneux et des Grandes Cultures (LBLGC), Université d'Orléans, INRA, USC1328, 45067 Orléans, France. [4] Department of Life Sciences, Imperial College London, Exhibition Road, South Kensington, London SW7 2AZ, UK. Correspondence and requests for materials should be addressed to K.M.T. (email: kathleen.trybus@uvm.edu) or to A.H. (email: anne.houdusse@curie.fr)

Malaria is a mosquito-borne disease caused by obligate protozoan parasites from the genus *Plasmodium*. Despite global efforts to control the disease, malaria is still responsible for half a million deaths each year, with the vast majority of mortality caused by *P. falciparum*[1]. During its complex life-cycle, the parasite transforms through motile and non-motile stages in its two hosts: human and mosquito (Supplementary Fig. 1a). The sporozoite stage that establishes infection, is injected into the dermis following the bite of an infected female mosquito from the genus *Anopheles*. This elongated and highly motile parasite form reaches dermal capillaries and is rapidly transported to the liver, where it targets and infects hepatocytes. Within hepatocytes the parasite grows and divides, eventually releasing tens of thousands of merozoite forms en masse into the blood stream. Merozoites, in contrast to sporozoites, are relatively non-motile stages that specifically target erythrocytes, where they develop and lead to all symptoms associated with malaria disease. Beyond this asexual replicative stage of development, some parasites switch commitment to develop into sexual forms (following an as yet unknown signal) producing male and female gametocytes that re-establish mosquito infection on the next bite[2] (Supplementary Fig. 1a). Whilst anti-malarials are available, resistance to all frontline drugs, including combination therapies based on artemisinin, continues to emerge[3] (Supplementary Fig. 1a), reinforcing the need to find drugs with new modes of action.

Force production and movement are essential for the progression at each stage of the parasite's two host life-cycle. Erythrocytic invasion by a merozoite, for example, involves a force of ~40 pN[4], while sporozoites can move at a speed of 2 μm.s$^{-1}$, an order of magnitude faster than the fastest human immune cells[5]. Like all Apicomplexa (the phylum to which *Plasmodium* belong), malaria parasites move using a substrate-dependent mechanism called gliding motility, a sophisticated substrate-dependent form of cell movement based on a macromolecular complex called the glideosome (Supplementary Fig. 1b). At the glideosome core is the single-headed class XIV myosin A (PfMyoA), and short and oriented filaments of a divergent actin (PfAct1). PfMyoA is a tailless myosin consisting only of a motor domain and light chain binding domain. The N-terminal region of one of the light chains (LC), called myosin tail interacting protein (MTIP), is believed to anchor the myosin to a membrane bound complex of glideosome associated proteins (GAP45-GAP50-GAP40) (Supplementary Fig. 1b). Several studies have investigated the role of myosin A in both the coccidian parasite *Toxoplasma gondii* and across the genus *Plasmodium*. In *Toxoplasma gondii*, Myosin A (TgMyoA) has been implicated in both invasion and egress of the parasite from the infected cell[6,7]. TgMyoA is not, however, essential for invasion and its function can be compensated for by TgMyoB or its splicing isoform TgMyoC[8,9], or by active host cell-mediated internalization[10]. Although host cell membrane wrapping forces likely play a role in merozoite invasion of red blood cells[11], their role has not been tested in the absence of a parasite motor. In the *Plasmodium* genus, initial experiments using the mouse malaria parasite *P. berghei*, showed that PbMyoA is required for cell motility and midgut colonization during parasite mosquito stages[12]. More recently, genetic ablation of GAP45 in *P. falciparum* blocked parasite invasion in merozoite stages, but not egress[13]. Whilst this study points to a critical role for the motor complex in blood stages of infection, the structural role GAP45 plays in apicomplexan cell architecture and glideosome function leaves the essential functionality of MyoA in the merozoite unresolved.

Given that the core of the glideosome is composed of divergent forms of both myosin and actin, the actomyosin system of *P. falciparum* is an attractive target for generation of new anti-malarial drugs. Actin is a highly conserved, ubiquitous protein in all Eukaryotes, with yeast and human actins sharing 87% identity, in contrast to <80% identity of PfAct1 with canonical actins. The sequence of PfAct1 differs from canonical actins in regions implicated in inter-protomer interfaces of the actin filament[14]. Accordingly, high resolution cryo-electron microscopy (CryoEM) studies confirmed altered inter-strand and intra-strand contacts that provided a structural basis for the instability of PfAct1 filaments[14,15]. PfMyoA is also a divergent motor, sharing only 30% sequence identity with Class II myosins. Interestingly, important sequence differences are located precisely in structural elements predicted to be relaying parts of the allosteric communication pathway within the myosin motor domain. In Class XIV myosins, to which PfMyoA belongs, the sequence of critical canonical residues for the motor mechanism are not conserved, including a pivotal $^{SH2-SH1}$Gly (G695 in scallop myosin 2) at the beginning of the SH1-helix. The connectors (Switch-2, the Relay helix, and the SH1-helix) are structural elements that coordinate rearrangements of the four motor subdomains, whose motions are amplified into a larger swing of the distal light chain binding lever arm (Supplementary Fig. 2). Mutations of this $^{SH2-SH1}$Gly greatly impede motor activity in class 2 myosins (Myo2) by reducing the flexibility of the fulcrum[16–18]. Despite lacking this fulcrum, PfMyoA nonetheless displays robust motor activity[19]. Resolving this conundrum requires structural studies of MyoA in different nucleotide states to reveal the key compensatory intra-molecular rearrangements that are involved in guiding the lever arm. Such a force generation mechanism is likely to be unique to the apicomplexan class XIV myosins. Based on a recently solved crystal structure of only one state (Pre-powerstroke, PPS) of the TgMyoA motor domain with a disordered N-terminal region, Boulanger and coworkers could only speculate about how sequence adaptations in the Wedge, the Relay and the fulcrum (Supplementary Fig. 2) may compensate for this mechanism[20].

By combining parasitological studies of MyoA in *P. falciparum* with structural, motility, and kinetic studies of the motor in vitro, we show that PfMyoA is critical for *P. falciparum* erythrocyte invasion, and that this atypical motor diverges significantly with regard to how force and motion is produced compared with conventional myosins. By solving three structural states of the PfMyoA motor domain, corresponding to three states of the motor cycle, we show how the unique class XIV N-terminal heavy chain extension and its interactions with other structural elements define motor function. We also show that Ser19 phosphorylation in this N-terminal extension substantially modulates the speed and force output generated by the motor. These insights allow us to propose a phosphorylation-dependent mechanism that tunes the motor for optimal invasion (high force) or dissemination (fast motility speed). Together, these insights provide a complete foundation from which to understand the non-canonical mechanism of force production in these globally significant parasites.

## Results

**PfMyoA is critical for erythrocytic invasion by merozoites.** Whilst GAP45 has been reported to be essential for erythrocytic invasion by merozoites[13] the role of the gliding motor itself, PfMyoA, in *Plasmodium* blood stages, has never been directly tested. To address this question, we generated a rapamycin (RAP) inducible PfMyoA KO[21] (Fig. 1a). Rapamycin treatment induces a constitutively expressed DiCre recombinase, leading to excision of the sequence between two *loxP* sites integrated into the *pfmyoa* gene[22], deleting the last 543 bases and bringing *gfp* into frame as a truncated fusion protein (residues 1–638). The resulting truncated PfMyoA motor lacks the C-terminal residues predicted

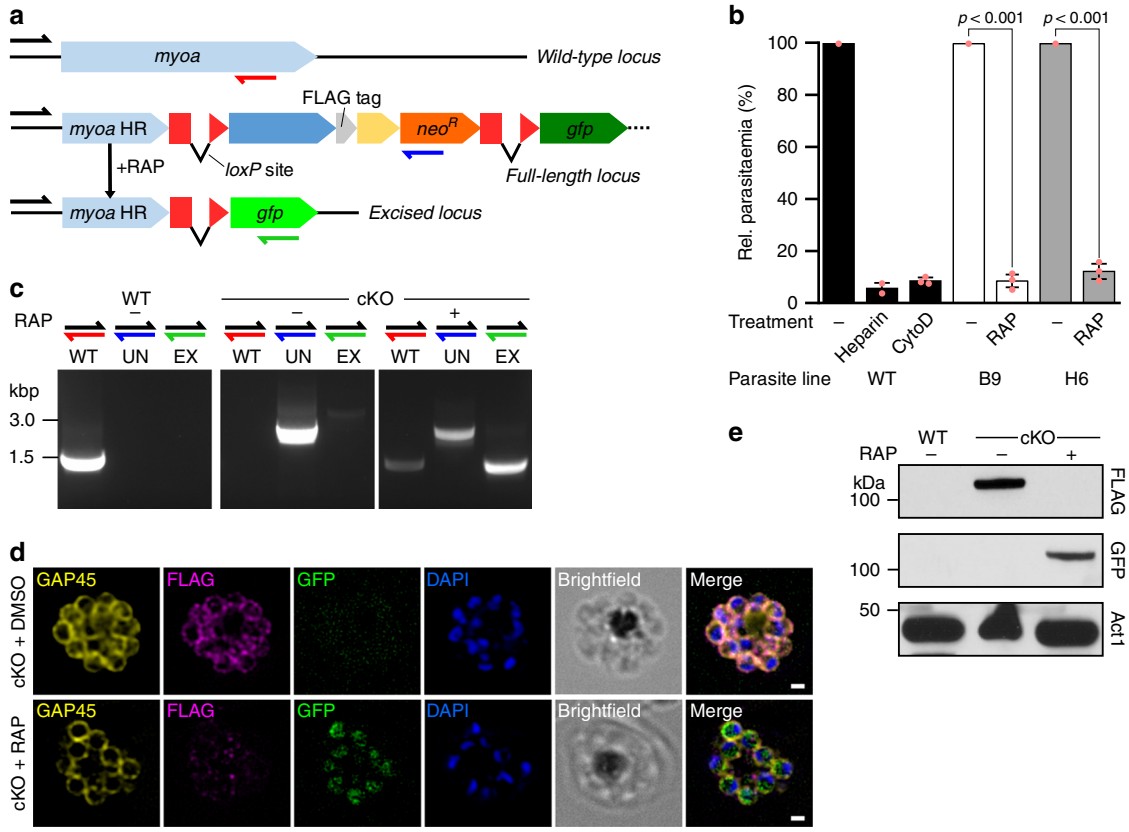

**Fig. 1** PfMyoA is critical for red blood cell invasion by merozoites. **a** Schematic showing replacement of the wild-type *myoa* locus with the full-length mutant locus by single crossover recombination in the *myoa HR* region using a T2A skip peptide (yellow box) to couple genomic integration to neomycin selection. **b** Treatment of two MyoA-cKO clones (B9 and H6) or WT with rapamycin (RAP) shows an almost complete invasion block (~90%) compared to DMSO treatment (−). This is comparable to the defect with known invasion inhibitors heparin and Cytochalasin D (CytoD) (90–95%). Parasitaemia was measured in the following cycle (cycle 1) by flow cytometry as the percentage of red blood cells (RBCs) that were DNA-positive by staining with SYBR Green I. Mean of three biological replicates (except for WT+heparin: two biological replicates), each three technical replicates, ± standard deviation (S.D.) of biological replicates. Data from each biological replicate were normalized to the DMSO-treated sample for each parasite line. Significance assessed using parametric t-test (paired, two-tailed). **c** Genotyping PCR of WT or cKO parasites following DMSO (−) or RAP (+) treatment detecting the wild-type (WT, red half-arrow), unexcised (UN, blue half-arrow) and excised (EX, green half-arrow) *pfmyoa* loci. **d** Immunofluorescence analysis of cKO parasites following DMSO or RAP treatment. FLAG-tag is detectable in DMSO-treated schizonts fixed ~48 h post-treatment, colocalising with motor complex protein GAP45, while after RAP treatment GFP but not FLAG is detectable, and the signal is restricted to the cytosol, consistent with a non-functional truncated MyoA. Scale bar 1 μm. Image stacks were deconvolved using the EpiDEMIC plugin for Icy, with a *z*-step size of 200 nm. **e** Western blot of WT and cKO parasites following DMSO (−) or RAP (+) treatment. Parasites were lysed ~40 h post-treatment, before the end of cycle 0. In DMSO-treated cKO parasites, the original FLAG-tag is detectable, but following RAP-treatment, only GFP is detectable. PfAct1 was used as a loading control. Representative blot shown from three biological replicates

to be required for the folding of the motor domain and would therefore be expected to be non-functional.

Two PfMyoA-cKO clones (B9 and H6) were compared with both WT and non-excised (DMSO treatment) controls to assess motor function. In addition, two known invasion inhibitors, heparin (that blocks cell-to-cell interactions) and CytoD (that blocks actin polymerization) served as positive controls[23] (Fig. 1b). Synchronized parasites were treated with 100 nM RAP and incubated for 16 h at the start of one cycle (cycle 0) to trigger truncation of *pfmyoa*. These were then tested for invasion of red blood cells (RBCs). Parasitaemia was measured in the following cycle (cycle 1) by flow cytometry as the percentage of RBCs that were DNA-positive by staining with SYBR Green I (Fig. 1b). Successful excision of the *loxP* section of *pfmyoa* was monitored by PCR (Fig. 1c), localization of PfMyoA by immunofluorescence (Fig. 1d) and western blotting (Fig. 1e), all confirming loss of the functional protein.

The KO of PfMyoA induced in the two PfMyoA-cKO clones demonstrated an almost complete block in invasion (~90%)

compared to DMSO treatment, comparable to rates of invasion detected with known invasion inhibitors heparin and CytoD (90–95%) (Fig. 1b). This confirms the long-held assumption that PfMyoA is directly involved in merozoite invasion of the erythrocyte in *P. falciparum*. Given the combined evidence that MyoA is critical for both invasion and for motility[12] in *Plasmodium* parasites, it would appear to be a bona fide first-order therapeutic target for preventing malaria parasite infection and disease progression.

**X-ray structures of three structural states of PfMyoA.** To define the motor cycle of PfMyoA, we determined the crystallographic structures of the motor domain (MD) of PfMyoA (1–768) in a nucleotide-free (NF) condition at 2.82 Å resolution (crystal type 1) (Supplementary Table 1, PDB code 6I7D) and complexed to ADP.Vanadate at 3.45 Å resolution (crystal type 2) (Supplementary Table 2, PDB code 6I7E). Four molecules are present in the asymmetric unit of crystal type 1, one of the motor domain is

in the Rigor-like state and three molecules adopt the Post-rigor (PR) state. These two structural states correspond, respectively to the state of highest affinity for F-actin[24,25] (Rigor-like) and to the state that allows detachment from F-actin but prior to the lever arm recovery stroke[26] (Post-rigor). The PfMyoA Rigor-like state adopts a fully closed cleft as found for non muscle Myo2c and Myo1b Rigor states[25,27] and is thus likely a good model of myosin XIV in the Rigor state.

The crystal type 2 contains one molecule per asymmetric unit. It corresponds to the Pre-powerstroke (PPS) state, a state with the lever arm up and able to dock to actin with electrostatic contacts. The resolution of the PPS state is poorer, but the electron density map is of good quality and allowed the entire motor domain to be built, including loops with good refinement statistics (Supplementary Table 2). Description of these three structural states for the PfMyoA motor provides a direct way to assess the major conformational changes that would occur during the motor cycle (Fig. 2a).

These structures enable visualization of the large conformational changes required for the powerstroke (PPS/Rigor), for detachment of the motor from actin (Rigor/PR), and for re-priming of the lever arm (PR/PPS). The amplitudes of the subdomain movements and converter swing are similar to that of other myosins[28] (Fig. 2a), but the stabilization of the states and the connector rearrangements differ. A key observation we made is the role of the unusual N-terminal (N-term) extension of PfMyoA (Fig. 2b, Supplementary Movie 1). This extension makes conserved interactions with the N-term subdomain in all three structures, and consists of a helix followed by a linker that contains the phosphorylated Ser19 (SEP19) (Supplementary Fig. 3). Structural analysis presented in detail below shows an unforeseen compensatory mechanism that involves the N-term extension driving the PfMyoA powerstroke despite the lack of sequence conservation in canonical residues of the motor domain connectors (Fig. 3a).

**Motor mechanism in classical myosins.** During motor domain rearrangements required for the motor cycle, movement near the active site involves the connector Switch-2, and is transmitted to the lever arm through interactions between Switch-2 and an important structural element of the Lower 50 kDa (L50) subdomain called the Wedge (Fig. 3b, Supplementary Fig. 2, 4). To drive the lever arm swing in all myosins studied thus far (classical myosins), these Switch-2 rearrangements are coordinated with specific changes of two other connectors: the Relay helix kink straightening and a SH1-helix rotation[28] (Fig. 3c, Supplementary Fig. 4, Supplementary Movie 2). The highly conserved glycine ($^{SH2-SH1}$Gly) acts as a fulcrum to promote the piston-like movement of the SH1-helix[29]. Conserved interactions between the Relay/SH1-helix and the Wedge/Switch-2 are critical for motor activity (Fig. 3, Supplementary Fig. 5, Supplementary Movies 2 and 3). In classical myosins, such as scallop myosin II (ScMyo2) (Supplementary Fig. 5a), the active site rearrangements

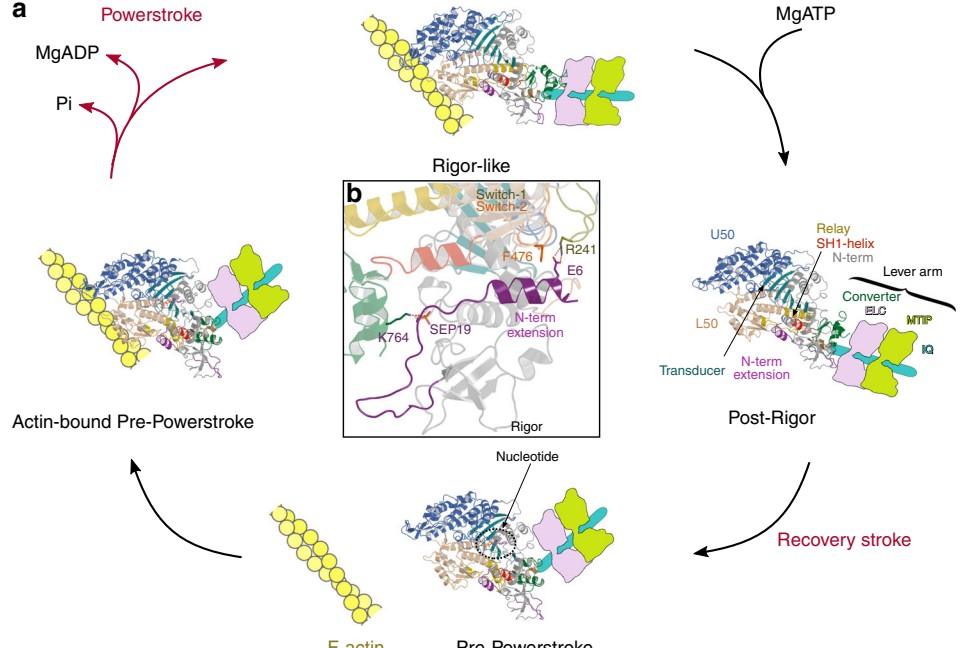

**Fig. 2** Structural states and motor cycle of PfMyoA. **a** The crystallographic structures of the three states of the motor cycle of PfMyoA are represented along the motor cycle: the Rigor-like and the Post-rigor (PR) states reveal how the motor detaches from F-actin upon ATP binding; the Pre-powerstroke (PPS) state corresponds to the state in which hydrolysis occurs and that rebinds to F-actin to trigger the powerstroke. The Rigor-like state is the conformation the motor adopts at the end of the powerstroke when hydrolysis products have been released. To appreciate the movement of the converter and how it can be amplified by the rest of the lever arm, the IQ region and the two LCs (PfELC and MTIP, Supplementary Fig. 1) are represented schematically in continuity of the last helix of the converter. PfMyoA displays the four canonical subdomains which are the hallmark of the myosin superfamily: N-terminus (N-term) (gray), Upper 50 kDa (U50) (marine blue), Lower 50 kDa (L50) (tint) and the converter (green) and central elements (including a beta-sheet) forming the transducer (dark cyan). During the motor cycle, rearrangements in the motor domain are allosterically transmitted through the Relay helix (yellow) and are amplified by the swing of the converter with the rest of the lever arm. **b** In the center diagram, a zoom of the Rigor-like structural state is presented to show the opposite side of the motor domain compared to **a**. Note the position of the unique N-terminal extension (purple) which is close to the connectors that direct rearrangements between motor subdomains (Switch-2 (orange), Relay (yellow) and SH1-helix (red)). PfMyoA employs an atypical motor mechanism in which the N-term extension (purple) compensates for non-canonical sequences in subdomain connectors essential for motor function

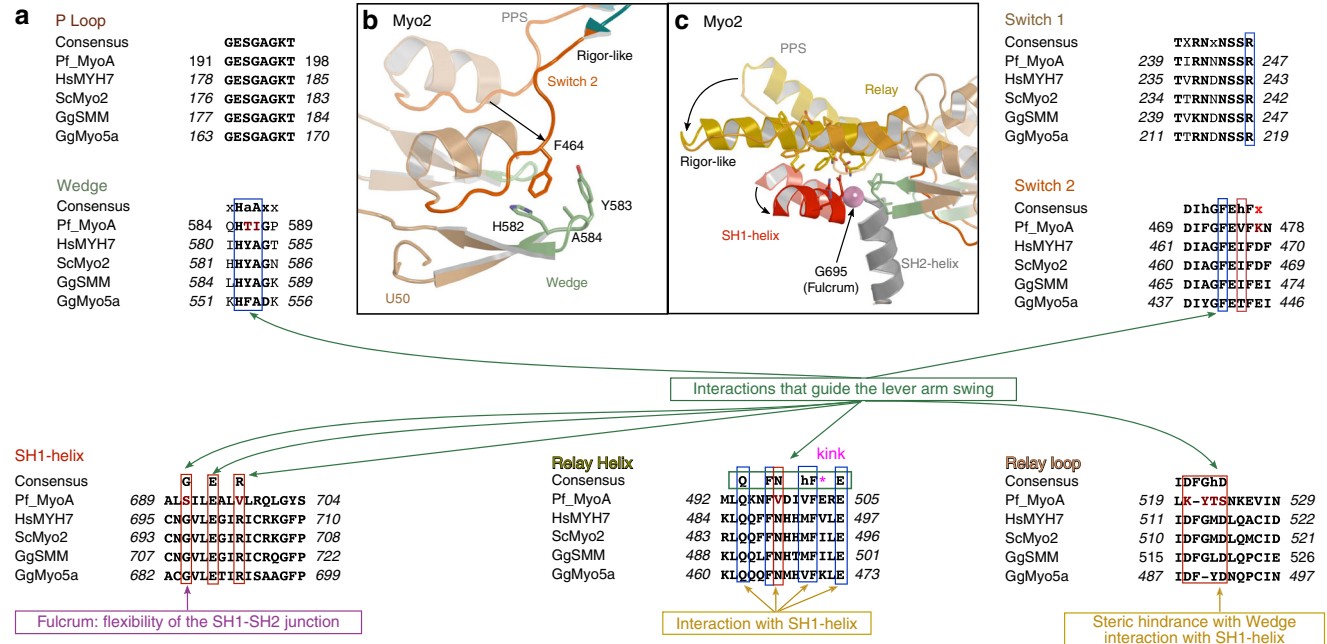

**Fig. 3** Sequence alignment of connectors essential in driving motor conformational changes–lack of canonical residues in PfMyoA. **a** Sequence comparison of several key elements involved in myosin mechanical transduction and allosteric communication. The consensus is represented on top of each sequence and the PfMyoA sequence is compared to other myosins. This comparison shows that the sequences of the Switch-2, Wedge, and Relay differ from the consensus. Consensus code: a is an aromatic residue, h a hydrophobic residue, x colored in red is an acidic residue. Residues important in allosteric communication in the motor are indicated in rectangles (blue rectangle if the residue is involved in a conserved interface, red rectangle if the residues are involved in interactions specific to PfMyoA). If the position is not conserved in PfMyoA, the residue is colored in dark red. To see how these residues affect the PfMyoA motor rearrangements along the cycle, see Fig. 4 and Supplementary Movie 2. **b** Switch-2/Wedge interactions in classical Myo2. **c** Conformational changes of two connectors before (PPS, transparent) and after (Rigor-like, plain) the powerstroke in Myo2. The Relay helix unkinks while the SH1-helix undergoes a piston-like movement using the canonical flexibility found at the fulcrum $^{SH2-SH1}$Gly (pink ball)

are linked to Switch-2 changes that are guided by hydrophobic interactions of $^{Switch-2}$F464 with canonical residues of the Wedge such as $^{Wedge}$H582 and $^{Wedge}$A584 (Supplementary Fig. 5b). In the PPS state, the Relay, the SH1-helix and the lever arm are in a "primed" conformation: the interactions between the Relay and the SH1-helix are mainly hydrophobic. Here the conserved $^{Relay}$M514 and $^{SH1-helix}$E698 are nearby (Supplementary Fig. 5b). During the powerstroke, release of hydrolysis products induces specific structural rearrangements in the active site, and specifically of Switch-2 that repositions the L50 subdomain. In so doing, the Wedge which belongs to the L50 subdomain becomes closer to the Relay/SH1-helix position. The steric hindrance caused by a conserved bulky aromatic, $^{Wedge}$Y583 with residues from the SH1-helix and the Relay, $^{Relay}$M514 and $^{SH1-helix}$E698, drives the straightening of the Relay, the piston-like movement of the SH1-helix (Supplementary Fig. 5b, Supplementary Movies 2 & 3) and thus the lever arm swings because these two connectors interact strongly with the converter (Supplementary Fig. 5c). The movement of the SH1-helix and the Relay are defined by interactions between these two connectors which are not only hydrophobic but supplemented with electrostatic interactions between conserved residues in the Rigor-like state: the conserved $^{SH1-helix}$R701 binds $^{Relay}$D515; $^{SH1-helix}$E698 binds $^{Relay}$Q485; and $^{Relay}$N489 (Supplementary Fig. 5b).

**The atypical structural motor mechanism of PfMyoA.** In PfMyoA, the amplitude of the converter swing is similar to that observed for classical myosins, despite the absence of the canonical residues essential for allosteric movement in classical myosins. This occurs while the SH1-helix is mostly immobile (Fig. 4), rather than performing a piston-like movement previously described for

classical myosins (Supplementary Fig. 5, Supplementary Movies 2 and 3). As predicted from sequence alignments, the replacement of the conserved $^{SH2-SH1}$Glycine restrains the mobility of the SH1-helix. Several sequence adaptations are needed to compensate for the lack of mobility of the SH1-helix. First, the aromatic and bulky residue of the Wedge is absent in PfMyoA and replaced by a threonine (T586), and the nearby bulky methionine residue from the Relay ($^{Relay}$511-DFGMD-515 in ScMyo2) is also absent and replaced by a threonine (T522) in PfMyoA ($^{Relay}$520-KYTS-523) (compare Fig. 4b and Supplementary Fig. 5b; Supplementary Movie 4). The movement of the Wedge during the powerstroke results in less steric hindrance and is thus adapted to the lack of mobility of the SH1-helix which stays in position while the Relay kink resolves and the lever arm swings (Fig. 4b, c, Supplementary Movie 3). Second, the interaction between the Relay and the SH1-helix is mainly hydrophobic both in the PPS state and in the Rigor-like state, with the exception of an electrostatic bond between the $^{Relay}$Q494 and the fulcrum $^{SH2-SH1}$S691 that is present in the two states (Fig. 4b, Supplementary Movie 3). Third, the PfMyoA Rigor-like state is stabilized by atypical interactions of $^{Wedge}$I587 with $^{Switch-2}$F473, $^{Switch-2}$V475, $^{SH2-helix}$H688 and $^{SH2-SH1}$S691 (red circle in Fig. 4b). Additional interactions with the N-term extension further stabilize the Rigor-like state, including a strong salt bridge-π interaction between the $^{N-term.extension}$Glu6, $^{Switch-1}$Arg241 and $^{Switch-2}$Phe476 (Fig. 4c). In addition, electrostatic contacts directly stabilize the converter since $^{Converter}$K764 interacts with the phosphorylated $^{N-term.extension}$SEP19 (Fig. 4c).

We conclude from the structures that the electrostatic interactions with the N-term extension compensate for the absence of a piston-like movement of the SH1-helix, enabling force production by accelerating the transitions of the

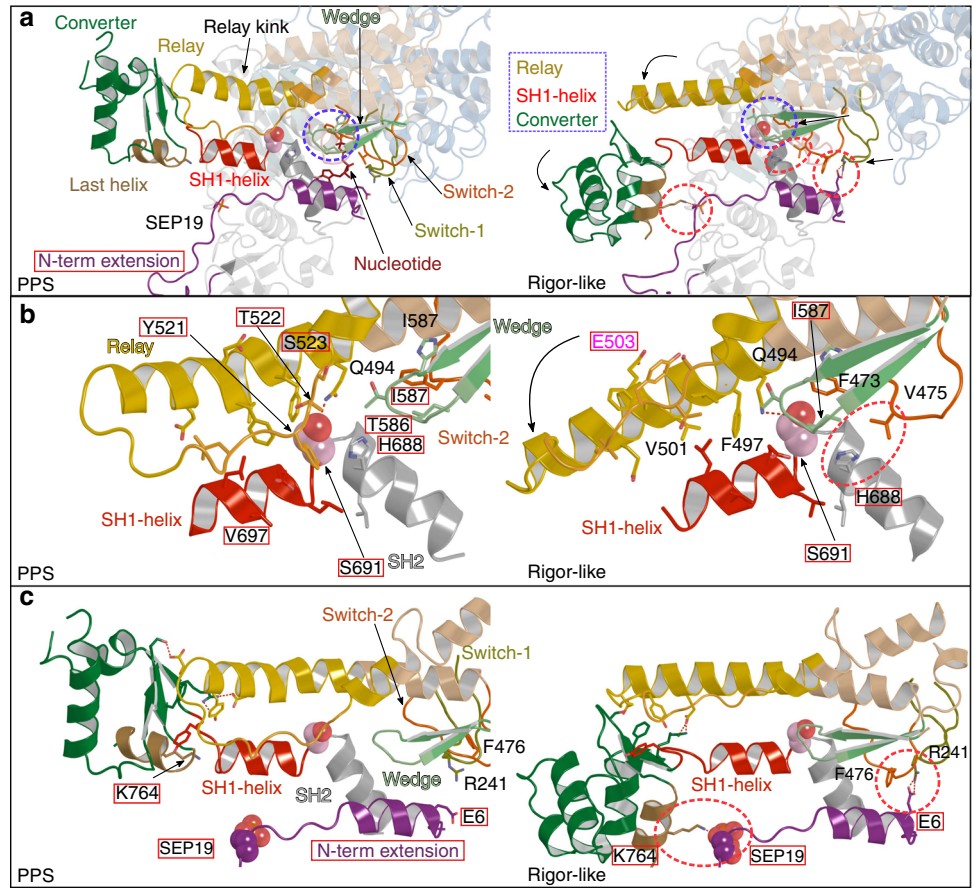

**Fig. 4** The unconventional mechanism of force production by PfMyoA. **a** Overall view of the mechanical communication within the PfMyoA motor domain during the powerstroke. In both the PPS and the Rigor-like states, interactions between Switch-2 and the Wedge are maintained (highlighted by dashed blue lines and detailed in **b**). The Rigor-like structure indicates that the sequential release of hydrolysis products upon the powerstroke triggers displacement of the Wedge which is associated with straightening of the Relay helix and the converter swing. The PfMyoA unconventional powerstroke requires sequence compensation near the Ser691 (blue dashed lines) since this residue is bulkier than the canonical [SH2-SH1]Gly found in classical myosins at this position. Thus, motor domain rearrangements are allowed by changes in the interactions between the Wedge and the Relay and SH1-helix connectors (details in **b**). Additional interactions (highlighted by dashed red lines) involving the N-term extension (purple) also stabilize the Rigor-like state and compensate for the immobility of the SH1-helix. (Details are shown using the same view in **c**). **b** Non-conserved residues are highlighted by a red rectangle. The SH1-helix lacks the conserved glycine [SH2-SH1]Gly at the fulcrum which is replaced by a serine (light pink spheres, S691). A hydrogen bond is formed between S691 and [Relay]Q494 in both the PPS and Rigor-like states. The presence of a less pliant fulcrum requires sequence adaptation in the Wedge and in the Relay and results in the immobility of the SH1-helix during the powerstroke. [Switch-2]V475 establishes hydrophobic interactions with [SH2]H688, helping to stabilize the Rigor-like position of the Switch-2 (red dashed lines). E503 (shown in pink) is a reporter to indicate the kink of the Relay. **c** In PfMyoA, the converter establishes a network of interactions with the Relay and the SH1-helix. Non-conserved residues are highlighted by a red rectangle. An electrostatic bond between phosphoserine 19 (SEP19) (N-term extension) and K764 (converter), as well as a salt bridge-π interaction between the [N-term extension]E6, [Switch-1]R241, [Switch-2]F476 (red dashed lines) stabilize the position of the converter in the Rigor-like conformation. For comparison with Myo2, see Supplementary Fig. 5 and Supplementary Movies 2 and 3

powerstroke. In particular, these interactions could accelerate ADP release which was slowed when the [SH2-SH1]Gly was mutated in a canonical Myo2 motor[17,18].

**PfMyoA phosphorylation tunes its motor properties**. To further analyze the compensatory mechanism and more fully decipher the role of the N-term phosphorylation, we functionally characterized PfMyoA constructs with a 19 amino acid N-term deletion (ΔN), or with point mutations that disrupt the N-term extension/converter interaction formed in the Rigor-like state (phospho-null S19A or K764E) (Figs. 4c, 5a). In agreement with the phosphorylated Ser (SEP19) observed in the crystallographic structures described above, we verified that PfMyoA was phosphorylated at Ser19 on the heavy chain during expression in *Sf*9 cells. Phosphoprotein gels qualitatively showed heavy chain phosphorylation, and mass spectrometry confirmed that 96 ± 1%

(± SD, n = 6, 2 independent preparations) of the peptides found in the untreated sample were phosphorylated compared with a phosphatase-treated sample (Supplementary Fig. 6).

In vitro motility assays showed that the N-term extension and, specifically the Ser19 phosphorylation, are necessary for actin displacement at maximal speed (~4 µm.s⁻¹). ΔN-PfMyoA reduced speed ~17-fold, while both the phospho-null (S19A) and charge reversal mutation in the converter (K764E) mutants slowed speed 2-fold to ~2 µm.s⁻¹ (Fig. 5b, Supplementary Table 3). The phosphomimic S19E did not fully recapitulate the enhancing effect of bona fide phosphorylation on speed, moving actin at a value intermediate (~2.8 µm/s) between S19A and phosphorylated PfMyoA (Supplementary Fig. 7a).

The same pattern seen for in vitro motility was also observed for the maximal rate of the steady-state actin-activated ATPase activity (WT >S19A or K764E >ΔN) (Fig. 5c). A linked assay with

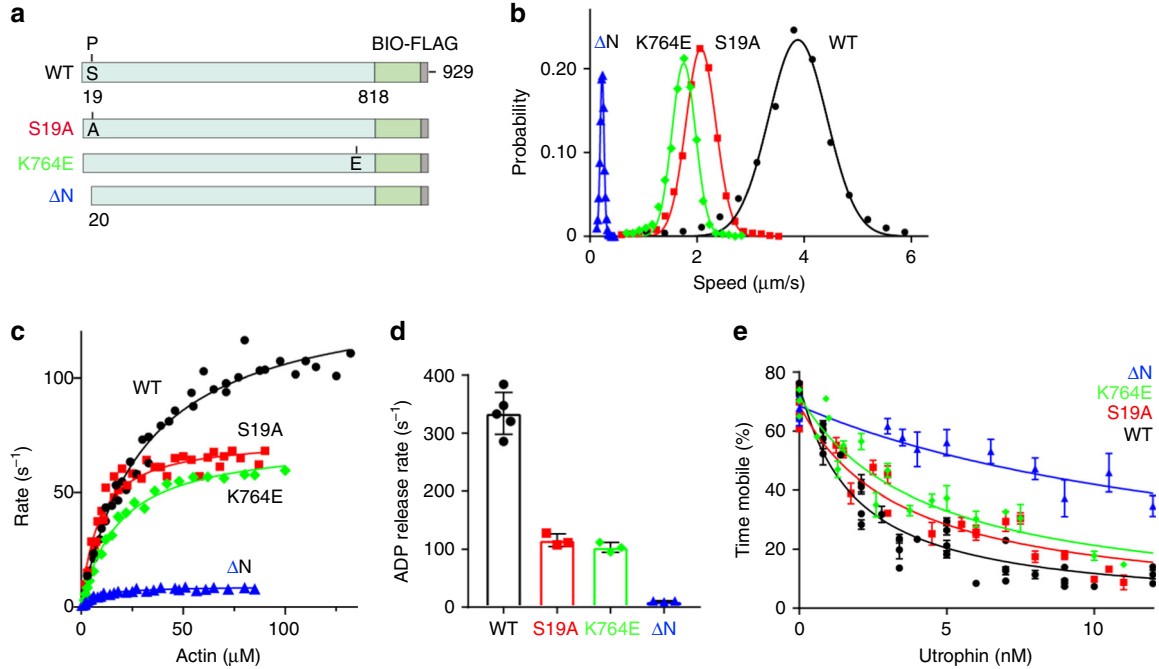

**Fig. 5** Functional properties of wild-type (WT) and mutant full-length PfMyoA constructs. **a** Schematic of constructs used for functional analysis. Ser19 is fully phosphorylated during protein expression in *Sf*9 cells (Supplementary Fig. 6). **b** Speed distributions from a representative in vitro motility assay. WT, $3.88 \pm 0.54$ µm/s; S19A, $2.07 \pm 0.28$ µm/s; K764E, $1.75 \pm 0.22$ µm/s; and ΔN, $0.23 \pm 0.04$ µm/s. Values, mean ± SD (Supplementary Table 3 shows data from multiple protein preparations). **c** Actin-activated ATPase activity for WT, $V_{max} = 138 \pm 4$ s$^{-1}$ and $K_m = 30.3 \pm 2.3$ µM; S19A, $V_{max} = 74.0 \pm 2.0$ s$^{-1}$ and $K_m = 8.5 \pm 1.0$ µM; K764E, $V_{max} = 72.9 \pm 1.9$ s$^{-1}$ and $K_m = 18.2 \pm 1.4$ µM; and ΔN, $V_{max} = 9.13 \pm 0.20$ s$^{-1}$ and $K_m = 7.34 \pm 0.67$ µM. Data from 2 protein preparations and 3 experiments for each construct were fitted to the Michaelis-Menten equation. Error, SE of the fit. **d** ADP release rates from acto-PfMyoA. WT, $334 \pm 36$ s$^{-1}$; S19A, $115.8 \pm 10.9$ s$^{-1}$; K764E, $103.5 \pm 8.6$ s$^{-1}$; ΔN, $10.32 \pm 0.91$ s$^{-1}$. Values, mean ± SD. WT vs. any other construct. $p < 0.0001$; S19A vs. K764E, NS; S19A or K764E vs. ΔN, $p < 0.01$ (one way ANOVA followed by a Tukey's Honest Significant Difference post-hoc test). Data from at least 3 protein preparations of each construct at different temperatures are shown in Supplementary Table 4. **e** Ensemble force measurements using a utrophin-based loaded in vitro motility assay. A myosin that produces more force requires higher utrophin concentrations to arrest motion: WT, $1.40 \pm 0.08$ nM; S19A, $2.42 \pm 0.17$ nM; K764E, $3.04 \pm 0.30$ nM; ΔN, $10.8 \pm 0.8$ nM. Error, SE of the fit. Data from two protein preparations and three experiments for each construct. Supplementary Fig. 7b shows these force data and fits extended to higher utrophin concentrations. Supplementary Fig. 7c–e shows ΔN data shown with an expanded y-axis. Skeletal actin was used for all experiments. Temperature, 30 °C. Source data are provided as a Source Data file

an ATP regenerating system was used to prevent ADP buildup during the course of the assay, which could potentially slow the ATPase rate. Transient kinetics were used to directly measure the rate of ADP release from acto-PfMyoA. A high concentration of MgATP was mixed with acto-PfMyoA (MgADP). Under these conditions, the rate of dissociation of PfMyoA from actin, as measured by light scattering, is limited by the rate of ADP release from the motor (Fig. 5d, Supplementary Table 4). For phosphorylated WT, S19A, and K764E, the rate of ADP release was faster than the steady-state ATPase rate. Interestingly, deletion of the N-term extension caused ADP release to become the rate-limiting step in the cycle (~10 s$^{-1}$), a hallmark of high duty-ratio motors that spend the majority of their ATPase cycle in a state strongly bound to actin[30].

Ensemble force measurements using a utrophin-based loaded motility assay and automated filament-tracking software[31] showed that ΔN produced ~8-fold more force, and S19A and K764E produced ~2-fold more force than phosphorylated PfMyoA (Fig. 5e, Supplementary Fig. 7b). These results demonstrate that Ser19 phosphorylation, which has been shown to occur in vivo[32], likely tunes PfMyoA motor properties. The absence of Ser19 phosphorylation, or removal of the N-term extension, enhances ensemble force at the expense of speed, confirming the role of this atypical extension in force generation.

The electrostatic bond between SEP19 and $^{Converter}$K764 stabilizes the Rigor-like conformation while it is unlikely to form in high ADP affinity states. The rate of the last

transition of the powerstroke (ADP release) is thus faster when Ser19 is phosphorylated. Faster ADP/ATP exchange and motor detachment from F-actin leads to higher speed by accelerating motor turn-over. The role of SEP19 in this transition was further investigated in silico using the mutant K764E. Molecular dynamic simulations on a 60 ns time-course further confirm the role of this electrostatic bond for the stability of the converter position, by comparing phosphorylated WT and K764E mutant in the Rigor-like state (Supplementary Fig. 8). In the WT phosphorylated motor domain, the converter is maintained in its Rigor-like position throughout the simulation (Supplementary Fig. 8a, c, d, Supplementary Movie 5). Conversely, in the K764E mutant, the converter position is not maintained and progressively deviates from its initial position together with the Relay, while the SH1-helix stays immobile (Supplementary Fig. 8b, c, d, Supplementary Movie 6). The in silico and in vitro results confirm that the electrostatic interaction between SEP19 and K764 stabilizes the Rigor-like state and is important for modulating ensemble force and the speed at which PfMyoA moves actin.

## Discussion
The divergent force production mechanism of PfMyoA, which we now show underpins the breadth of *Plasmodium* parasite lifecycle progression, is the result of sequence adaptations in which the N-term extension compensates for the absence of the $^{SH1-SH2}$Gly fulcrum residue, shown as essential for the piston-like movement of the SH1-helix of canonical myosins[18]. By providing three

structural states of the PfMyoA motor, our data directly show the atypical mechanism of force production. While the sequence adaptation near the [SH1-SH2]Gly/Wedge was correctly predicted from the TgMyoA PPS structure as part of this mechanism[20], the authors could not envisage that this would lead to a lack of piston-like movement of the SH1-helix as shown here. Other speculations in the study were incorrect because of an inability to visualize the N-term extension in the TgMyoA structure and the absence of a Rigor-like state structure. The details provided by the PfMyoA structures here, highlight how this N-term extension plays a direct and critical role in the force generation mechanism by direct interactions with Switch-2 and the converter.

The PfMyoA Rigor-like structure reveals how an interaction of the phosphorylated serine in the N-term extension with the converter is directly involved in the last step of the lever arm swing associated with ADP release. In so doing, phosphorylation directly controls the speed and force sensitivity of the PfMyoA motor by modulating the time spent strongly attached to F-actin. These insights into the PfMyoA mechanical cycle reveal that a specific phosphorylation event on class XIV myosins tunes their motor activity by a change in their duty ratio. This allows switching between an ability of the motor to generate force under loaded conditions, or to move actin for highest speed under unloaded conditions. The ATPase activity and unloaded in vitro motility speed of expressed PfMyoA were also recently measured by Green et al.[33], but with values for both that were over 10-fold lower than reported here. Although part of this discrepancy (~2-fold) can be attributed to different assay temperatures (our study 30 °C, their study 23 °C), and part to not knowing the state of phosphorylation of their expressed myosin (dephosphorylation would slow values 2-fold), it is unclear why their values are considerably lower than reported here.

In mammalian Myo1b, ADP release has been shown to involve the N-terminal extension of the motor that docks at the motor domain/lever arm interface in the Rigor state but is free of interaction in the strong ADP state[27,34]. The position of the N-terminal extension and the location of the bonds that favor the Rigor-like state differ greatly between PfMyoA and Myo1b (Supplementary Fig. 10), although in both cases, the presence of the N-terminal extension increases ADP release by forming precise bonds stabilizing the Rigor state. No phosphorylation has been reported for the Myo1b N-terminal extension which contains two serine residues (Ser8, Ser9), so it is unknown whether the tuning of ADP release found by phosphorylation of PfMyoA might be recapitulated in other members of the myosin superfamily such as Myo1b. If phosphorylation on either of these Myo1b serines occurs, it would greatly perturb and likely prevent the docking of the N-terminal extension required for stabilizing the Rigor state. Thus, while both N-terminal extensions contribute to stabilizing the Rigor state, the nature of how this is done differs greatly between these myosins. The PfMyoA N-term extension, in contrast to that of Myo1b, is integral to the motor domain and directly influences the connectors that drive the lever arm swing in addition to the position of the converter. Depending on its phosphorylation state, the PfMyoA N-term extension stabilizes the Rigor state in a unique and tunable way to control the motor duty ratio.

The mechanism presented here provides new insights into how PfMyoA may modulate its motor properties and adapt for optimized Plasmodium parasite function across the lifecycle. Liver-stage sporozoites can move at speeds of >2 µm.s$^{-1}$ across substrates or within extra-cellular space. This high speed is one of the fastest characterized for any eukaryotic cell. In contrast, blood stages merozoites, which are relatively immotile away from host cells, require the development of force (~40 pN) by the glideosome against the erythrocytes specifically for invasion[4]. Because it is now clear that PfMyoA is required for both motility and

invasion processes, Ser19 phosphorylation could provide a mechanism to optimize motor properties (speed vs. force) depending on parasite life-cycle stage (Fig. 6). From proteomic studies there is evidence of S19 phosphorylation in schizonts, merozoites[32] and salivary gland sporozoites[35] but further study with invasion-ready parasites will be required to show the extent of S19 phosphorylation in each stage. Motility studies with liver-stage sporozoites would be required to definitively test this hypothesis and how it relates to the subtleties of motor tuning.

All the Apicomplexan parasites differ in their life-cycle, as well as in the host organisms that they target. The function of myosin A amongst these organisms may differ too, as suggested by the differences found for the role and essentiality of MyoA in the genus Plasmodium vs. Toxoplasma. In T. gondii, TgMyoA is essential for egress[6,7] but not for host cell invasion[8,9]. Here, we demonstrate that PfMyoA is critical for erythrocytic invasion in P. falciparum, but the parasite is still able to egress in the absence of GAP45, the protein anchoring PfMyoA to the inner membrane complex[13]. This example shows likely key differences in the cellular basis of host cell entry and exit across Apicomplexan parasites, highlighting that caution should be used when extrapolating conclusions from one genus to another.

These results also raise the question about the conservation of the atypical and tunable mechanism of PfMyoA in other apicomplexan parasites. All the sequence adaptations described here are present in the recently solved PPS structural state of TgMyoA[20]: the G-to-S substitution at the SH2-SH1 fulcrum; the sequence adaptation to maintain the interaction between the Relay and the SH1-helix; the substitution of the aromatic residue of the Wedge by a threonine. In the TgMyoA converter there is a lysine (K766) equivalent to K764 in PfMyoA. Moreover, the TgMyoA N-term extension contains three serines that can be phosphorylated: S20, S21 that is equivalent to S19 in PfMyoA, and S29. An additional serine can be phosphorylated (S743) in the converter. Phosphomimetic mutants showed that the number of charged residues in the N-terminus modulates the speed at which TgMyoA moved actin filaments[20], although we showed here that a phosphomimic does not recapitulate the full effect of a phosphorylated Ser. While the N-term extension was not defined in the electron density of the TgMyoA X-ray structure, the sequence conservation is sufficient to assume that it may form a helix and a turn, as found in our PfMyoA structures (Supplementary Fig. 3d, e). The basis of the tunable mechanism described here for PfMyoA is thus likely conserved in TgMyoA, although possibly more complex because it involves two additional phosphorylation sites in the N-term extension and one additional site in the converter. These differences in the possible tuning of PfMyoA and TgMyoA activity may result from variations of the requirement for the glideosome motor during the different life-cycles of the two parasites. Indeed, across its human host, Plasmodium falciparum alternates between the highly motile sporozoite and the invasive but relatively immobile merozoite. In contrast, the free stage of Toxoplasma gondii is the tachyzoite with a speed of 5 µm.s$^{-1}$, similar to the maximum in vitro motility speed for TgMyoA when this myosin had three N-terminal phospho-mimic Asp residues[20]. Tachyzoites, however, can either undergo dissemination, invasion or egress. Thus, controlling the phosphorylation status of TgMyoA in tachyzoites could in principle modulate the speed and/or force developed by TgMyoA and thus adapt its motor properties depending on the need for either the dissemination, invasion or egress phases of the parasite. It may be possible to test this hypothesis with future studies on TgMyoA, permitting a direct comparison to the PfMyoA tuning mechanism revealed here.

Finally, we show here definitively for the first time that PfMyoA is critical for erythrocytic invasion. Myosin A is thus not only

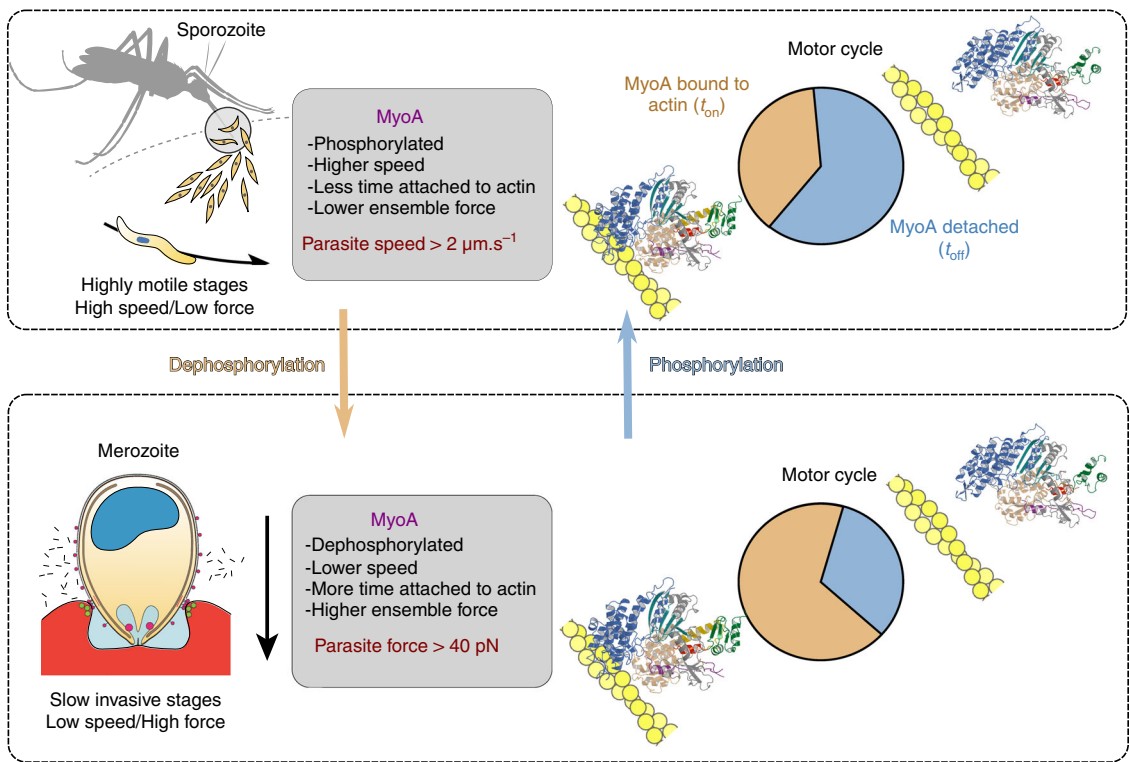

**Fig. 6** Phosphorylation of PfMyoA tunes its motor properties. Scheme representing how phosphorylation tunes PfMyoA motor properties and how this could optimize the motor for parasite motility or invasion at different stages of the parasite. In highly motile stages, sporozoites move at a speed higher than 2 µm/s. At this stage, phosphorylation of PfMyoA would allow the parasite to move actin at maximal speed but with low ensemble force. In merozoites, which lack continuous motility, but are instead adapted for erythrocyte invasion, dephosphorylation of PfMyoA localized at the invasion junction would result in active motors efficient in invasion. This merozoite PfMyoA motor would spend more of its total cycle time strongly bound to actin, thereby resulting in greater ensemble force output. The apparent duty ratio in the two phosphorylation states was estimated from the rate of ADP release divided by the total ATPase cycle time. The S19A, K764E and ΔN mutants would likely impair speed but not invasion, because their ensemble force is higher than phosphorylated, wild-type PfMyoA. Phosphorylation of the N-term extension of PfMyoA can thus act as a switch to tune motor activity depending on the needs of the parasite (high speed and low force for gliding, or higher force for the low speed invasion process)

required for motility in general in the *Plasmodium* genus but also for establishing blood stage infection during the symptomatic stages. The levels of reinvasion are commensurate with a complete block in invasion, as shown by comparison with CytoD and heparin. Nevertheless, there may be some residual invasion due to parasites with non-excised *pfmyoa* locus and there is a possibility of invasion by RBC membrane wrapping interactions alone[11]. However, the size of this residual population appears much smaller than found in *T. gondii* tachyzoites lacking TgMyoA[9] and would argue against redundancy in motor function like that seen in *Toxoplasma*.

In recent years, several small molecules targeting myosins have been developed as promising treatments for diverse pathologies such as cardiac diseases and asthma[36–39]. Amongst these modulators, CK-571 is able to selectively block smooth muscle myosin by trapping an intermediate of the recovery stroke[38]. The design of molecules able to block PfMyoA activity based on similar principles, such as that found for CK-571, would be a highly specific and efficient way to treat malaria. A ~1000-fold difference in IC50 allows CK-571 to be tuned for inhibition of smooth muscle myosin compared with cardiac or skeletal muscle myosins, despite the fact that the allosteric drug binding pocket corresponds to conserved residues in sequence among these Myo2 motors. The divergence of PfMyoA in terms of sequence and mechanism would ensure an even higher selectivity for inhibitors of the recovery stroke. The atypical mechanism of PfMyoA described here and the structural blueprints provided should therefore enable the specific design of inhibitors that could stall the progression of malarial infection.

## Methods

**Expression constructs**. Full-length PfMyoA heavy chain (PlasmoDB ID PF3D7_1342600/ GenBank accession number XM_001350111.1), with Sf9 cell preferred codons, was cloned into the baculovirus transfer vector pFastbac (pFB). A 13 amino acid linker separates the C-terminus of the PfMyoA heavy chain from an 88 amino acid segment of the *Escherichia coli* biotin carboxyl carrier protein[40], which gets biotinated during expression in Sf9 cells, followed by a C-terminal FLAG tag for purification via affinity chromatography. The S19A or K764E mutated versions were generated using site directed mutagenesis. The ΔN version of full length PfMyoA heavy chain was made by removing residues Ala2 through Ser19. All PCR products were cloned into the baculovirus transfer vector pAcSG2 (BD Biosciences) to make recombinant virus for the Sf9 expression system. The mouse utrophin (NP_035812) clone was a gift from Kathleen Ruppel and James Spudich. It was modified so that utrophin residues 1-H416 were followed by C-terminal biotin and FLAG tags. It was cloned into pFastbac for production of recombinant baculovirus and subsequent expression in Sf9 cells.

**Protein expression and purification**. The full-length PfMyoA heavy chain or mutant constructs (S19A, K764E, ΔN) were co-expressed with the chaperone PUNC and the light chains (PfMTIP and PfELC) in Sf9 cells as described in ref. [19]. The cells were grown for 72 h in medium containing 0.2 mg/ml biotin, harvested and lysed by sonication in 10 mM imidazole, pH 7.4, 0.2 M NaCl, 1 mM EGTA, 5 mM MgCl₂, 7% (w/v) sucrose, 2 mM DTT, 0.5 mM 4-(2-aminoethyl)benzene-suflonyl fluoride, 5 µg/ml leupeptin, 2 mM MgATP. An additional 2 mM MgATP was added prior to a clarifying spin at 200,000×g for 40 min. The supernatant was purified using FLAG-affinity chromatography (Sigma). The column was washed with 10 mM imidazole pH 7.4, 0.2 M NaCl, and 1 mM EGTA and the myosin eluted from the column using the same buffer plus 0.1 mg/ml FLAG peptide. The fractions containing myosin were pooled and concentrated using an Amicon centrifugal filter device (Millipore), and dialyzed overnight against 10 mM imidazole, pH 7.4, 0.2 M NaCl, 1 mM EGTA, 55% (v/v) glycerol, 1 mM DTT, and 1 µg/ml leupeptin and stored at −20 °C. Utrophin purification was essentially the same as for myosin but without the MgATP steps. Skeletal muscle actin was purified from chicken skeletal muscle tissue essentially as described in[41].

His-tagged PfMTIP and PfELC were bacterially expressed using BLR(DE3)-competent cells and the isopropyl $\beta$-D-thiogalactopyranoside induction system[40]. Pellets were lysed by sonication in 10 mM sodium phosphate, pH 7.4, 0.3 M NaCl, 0.5% (v/v) glycerol, 7% (w/v) sucrose, 7 mM $\beta$-mercaptoethanol, 0.5 mM 4-(2-aminoethyl)benzenesuflonyl fluoride, and 5 µg/ml leupeptin. The cell lysate was clarified at 26,000×g for 30 min. The supernatant was boiled for 10 min in a double boiler, clarified at 26,000×g for 30 min, and the supernatant loaded on a His-select nickel-affinity column (Sigma). The resin was washed in 10 mM sodium phosphate, pH 7.4, 0.3 M NaCl before being eluted in the same buffer containing 200 mM imidazole. The protein was concentrated and dialyzed overnight against 10 mM imidazole, pH 7.4, 150 mM NaCl, 1 mM EGTA, 1 mM MgCl$_2$, 55% (v/v) glycerol, 1 mM DTT and stored at −20 °C.

**Crystallization and data processing**. Crystals of PfMyoA motor domain in Post-rigor and Rigor-like states (type 1 crystals) (10 mg.ml$^{-1}$) were obtained at 4 °C by the hanging drop vapor diffusion method from a 1:1 mixture of protein with 2 mM EGTA and precipitant containing 18% PEG 3350; 100 mM glycine pH 8.6. Crystals of PfMyoA motor domain in Pre-powerstroke state (type 2 crystals) were obtained at 17 °C by the hanging drop vapor diffusion method from a 1:1 mixture of protein (21 mg.ml$^{-1}$) with 2 mM MgADP.VO$_4$ and precipitant containing 0.6 M K-Phosphate dibasic and 0.7 M Na-Phosphate monobasic. Crystals were transferred in the mother liquor containing 30% glycerol before flash freezing in liquid nitrogen. X-ray diffraction data were collected at 100 K on the PX1 ($\lambda = 0.906019$ Å) for type 1 crystals and on the PX2 ($\lambda = 0.979384$ Å) for type 2 crystals, at the SOLEIL synchrotron. Diffraction data were processed using the XDS package[42] and AutoProc[43]. Data obtained from the crystal type 2 were anisotropic, processing has been performed with StarAniso[44]. The crystals type 1 belong to the P2$_1$ space group with four molecules per asymmetric unit. The type 2 crystals belong to the P6$_1$22 space group with one molecule per asymmetric unit. The cell parameters and data collection statistics for these crystals are reported in Supplementary Tables 1 and 2, respectively.

**Structure determination and refinement**. Molecular replacement in type 1 crystals was performed with the motor domain (residues 66–688) of scallop myosin 2 in Post-rigor state (PDB code 1B7T) and in Rigor-like state (PDB code 2OS8) without water and ligand with Phaser[45] from the CCP4i program suite. Manual model building was achieved using Coot[46]. Refinement was performed with Buster[47]. The statistics for most favored, allowed and outlier Ramachandran angles are 95.63, 3.97, and 0.4%, respectively. Structure determination indicated that this crystal form corresponds to three molecules in the Post-rigor state (chains A, B, C) and one molecule in the Rigor-like state (chain D). The final model has been deposited on the PDB (PDB code 6I7D). The procedure for structure determination and refinement for crystals type 2 was identical, using one molecule of PfMyoA in Post-rigor state without the converter (residues 2–704) and without water and ligand as search model for molecular replacement. The statistics for most favored, allowed and outlier Ramachandran angles are 82.49, 13.40, and 4.11%, respectively. Structure determination indicated that this crystal form corresponds to the Pre-powerstroke state. The final model has been deposited on the PDB (PDB code 6I7E).

**Molecular dynamics simulations**. The procedure used for molecular dynamics simulation is similar to what was described in[48]. Inputs were prepared with CHARMM-GUI[49] with the Quick MD Simulator module[50]. The CHARMM36 force field[51] was used to describe the full systems in a box with explicit water (TIP3P model) and salt (KCl reaching 150 mM). The protocol provided in the Quick MD output was followed. Gromacs[52] (VERSION 5.0-rc1) was used to execute the 60 ns simulations. Each molecular dynamics simulation was analyzed with the Gromacs tools.

**Motor cycle reconstitution and visualization**. In this study, we reconstitute the motor cycle based on three structural states: Post-rigor (PR), Pre-powerstroke (PPS) and Rigor-like. To visualize structural changes during the powerstroke, we used a comparison between the PPS and the Rigor-like states. We have made some animations (Supplementary Movies 1–4) to compare the conformation of the different connectors in the PPS and in the Rigor-like in conventional myosins (ScMyo2) and in the atypical PfMyoA. Morphs were not used to illustrate the transition between the PPS and the Rigor-like states since the isomerization occurs through intermediate states[53] which are not known for these myosins, this kind of procedure would thus be misleading.

In order to visualize the powerstroke of a conventional class II myosin, we have used scallop myosin II (ScMyo2). The structures of the Pre-powerstroke (PPS) and of the Rigor-like states of ScMyo2 have been solved in different organisms: PPS of bay scallop (*Argopecten irradians*) ScMyo2 (PDB code 1QVI, resolution 2.54 Å) and Rigor-like state of Atlantic deep-sea scallop (*Placopecten magellanicus*) ScMyo2 (PDB code 2OS8, resolution 3.27 Å). The sequence of ScMyo2 heavy chain in these two species is highly conserved for the motor domain (92% identity, 95% similarity) and the two structures can thus be used to reconstitute the powerstroke of ScMyo2.

**Unloaded and loaded in vitro motility measurements**. Before the assay, myosin heads that cannot release from actin upon binding MgATP were removed by a 20 min spin at 350,000×g in the presence of 1.5 mM MgATP and a three-fold excess of skeletal actin. Myosin and utrophin (when applicable) concentrations were determined using the Bio-Rad Protein assay. The following solutions were added to a nitrocellulose-coated flow cell in 15 µl volumes. 0.5 mg/ml biotinylated bovine serum albumin (BSA) in buffer A (150 mM KCl, 25 mM imidazole pH 7.5, 1 mM EGTA, 4 mM MgCl$_2$, and 10 mM DTT) was added and incubated for 1 min. Three additions of 0.7 mg/ml BSA in buffer A and a 2 min wait prevent subsequent nonspecific binding of either the myosin or utrophin. 25 µg/ml neutravidin (Thermo Fischer Scientific) in buffer A was added for 1 min and then rinsed three times with buffer A. 120 µg/ml PfMyoA (and utrophin, when performing a loaded in vitro motility assay) was flowed into the cell and rinsed 3 times with buffer A after a 1 min wait. All myosin and utrophin constructs contained a C-terminal biotin tag for specific attachment to the neutravidin coated flow cell. Rhodamine-phalloidin labeled skeletal muscle actin was incubated for 30 s followed by one rinse with buffer A and one rinse with buffer B. Buffer B is buffer A plus 0.5% (w/v) methylcellulose, 25 µg/ml PfELC and 25 µg/ml PfMTIP and oxygen scavengers (50 µg/ml catalase (Sigma), 125 µg/ml glucose oxidase (Sigma), and 3 mg/ml glucose). Motility was initiated by adding buffer B containing 2 mM MgATP twice to the flow cell, and waiting 1 min for temperature equilibration (30 °C) on the microscope.

Actin filaments were visualized using an inverted microscope (Zeiss Axiovert 10) equipped with epifluorescence, a Rolera MGi Plus digital camera, and a dedicated computer with the Nikon NIS Elements software package. Filaments were automatically tracked and analyzed, both under unloaded and loaded conditions, with the Fast Automated Spud Trekker analysis program (FAST, Spudich laboratory, Stanford University). This online software is available free for download (http://spudlab.stanford.edu/fast-for-automatic-motility-measurements/).

Speeds for all unloaded motility experiments were fit to a Gaussian curve. All loaded motility data sets were fitted to the equation:

$$\text{Time mobile} = \frac{K_S M_0}{K_S + M_0 [\text{utrophin}]}$$

Where $M_o$ is percent mobile filaments in unloaded conditions, $K_S$ is a pseudo actin-utrophin dissociation constant (in nM), that is sensitive to myosin ensemble force, and % time mobile is the total duration of filaments in the mobile state divided by the total duration of the combined stuck and mobile states (see ref. [31] for a detailed explanation).

**Actin-activated ATPase activity**. The actin-activated ATPase activity of all PfMyoA constructs was determined at 30 °C in 10 mM imidazole pH 7.5, 5 mM KCl, 1 mM MgCl$_2$, 1 mM EGTA, 1 mM DTT, and 1 mM NaN$_3$ as a function of skeletal actin concentration. The salt concentration was kept low (5 mM) to minimize $K_m$ values and reach $V_{max}$ with actin concentrations that are compatible with this assay. Actin-activated ATPase activity was determined using a linked assay, which couples the regeneration of hydrolyzed ATP to the oxidation of NADH to NAD+. Pyruvate kinase (Sigma), in the presence of phophoenolpyruvate (Sigma) and low ADP concentrations, regenerates ATP. In a subsequent reaction, pyruvate is converted to lactate by L-lactate dehydrogenase (Sigma) which oxidizes NADH (Sigma) to NAD+. The decrease in optical density at 340 nm as a function of time was measured on a Lambda 25 UV/VIS spectrophotometer (Perkin Elmer), with a 300 s data monitoring window sampled every 2 s. Data were fit to the Michaelis–Menten equation.

**Transient kinetics**. All kinetic experiments were performed in 10 mM 4-(2-hydroxyethyl)-1-piperazineethanesulfonic acid, pH 7.5, 50 mM KCl, 4 mM MgCl$_2$, 1 mM EGTA, and 1 mM DTT at 30 °C using a KinTek stopped-flow apparatus (KinTek Corporation, model SF-2001). All concentrations stated are those after mixing in the stopped-flow cell. Light scattering at 295 nm was used to monitor the dissociation of the actomyosin.ADP complex (skeletal actin) by 2 mM MgATP. Light scattering of unlabeled acto-PfMyoA was observed with an incident beam of 295 nm and detected orthogonal to the incident beam with a 295 nm interference filter. All traces were analyzed using the software provided by KinTek and fit to single exponential fits. Typically, multiple (3–10) time courses were averaged before fitting to an exponential.

**Mass spectrometry**. The phosphorylation status of expressed PfMyoA was determined by liquid chromatography tandem mass spectrometry (LCMS/MS). Aliquots of PfMyoA heavy chain (104 kDa) and PfMyoA heavy chain treated with dephosphorylation with calf intestinal phosphatase (CIP; New England Biolabs, M0290S) were separated by SDS-PAGE, stained with Coomassie and excised from the gel. The bands were destained with 50% acetonitrile (ACN), dehydrated with 100% ACN and dried in a speed vacuum device. The bands were destained, reduced, alkylated and digested with 2 µg of trypsin (Promega) as described[19]. The resultant peptides were dried in a speed vacuum device and then reconstituted in 0.05% trifluoroacetic acid. The peptides were separated on an Acquity UPLC HSS T3 column (100 Å, 1.8 µm, 1 mm × 150 mm) (Waters) attached to a Dionex Ulti-Mate 3000 high pressure liquid chromatography system (HPLC) (Dionex). The

HPLC effluent was directly injected into a Q Exactive Hybrid Quadrupole-Orbitrap mass spectrometer through an electrospray ionization source (ThermoFisher). Data were collected in data-dependent MS/MS mode with the top 5 most abundant ions being selected for fragmentation.

Peptides were identified from the resultant MS/MS spectra using SEQUEST run via Proteome Discoverer 2.2 software (ThermoFisher). These searches were performed against a custom database that reflected the cloned gene (PfMyoA heavy chain) plus tags. Peptide oxidation was accounted for by addition of 15.99 and 31.99 Da to each methionine, carbamidomethylation was accounted for by addition of 57.02 Da to each cysteine, and phosphorylation was accounted for by adding 79.97 Da to each serine, threonine or tyrosine residue. All identifications were manually confirmed by inspection of the MS/MS fragmentation spectra. LC peak areas were quantified for each peptide using Proteome Discoverer 2.2. The LC peak areas were normalized using the median abundance of the top 5 most abundant PfMyoA peptides. The degree of phosphorylation was estimated from abundance of non-phosphorylated VSNVEAFDK peptide (containing serine 19) in the untreated PfMyoA samples relative to that in the samples dephosphorylated with CIP using a mass-balance approach[54].

**DNA manipulation.** For generation of Myo-LP constructs, synthetic gene fragments were ordered (GENEWIZ or Thermo Scientific) comprising the re-codon optimized myosin sequence, *loxPint* module, tags and T2A skip peptide with 30 bp Gibson flanks. These were assembled with a ~700 bp long homology region (amplified from 3D7 genomic DNA) by Gibson assembly. The parent vector (pARL-FIKK10.1) was a gift of Dr Moritz Treeck (Crick Institute). Plasmids were purified from *E. coli* cultures using QIAGEN Plasmid Maxi kit.

**P. falciparum culture and transfection.** Strain B11[13], which constitutively expresses inducible DiCre, was cultured in RPMI 1640 (Life Technologies) under standard conditions[55] at 4% hematocrit and synchronized with 5% sorbitol (Sigma)[56]. Parasites were grown to 5% at ring stage and electroporated with 100 μg of plasmid in 15 μL of sterile TE buffer added to 385 μL sterile Cytomix[57]. Plasmid uptake was selected for by adding fresh media with 3 nM WR99210 daily for 7 days, then every 2–3 days until parasite population re-established. Transfectants were grown to 2–4% parasitaemia and integration was selected for with 400 μg/mL G418[21]. Fresh media with G418 was added for 10 days before parasites were returned to drug-free media. MyoA-cKO parasites were cloned by limiting dilution.

**Genotyping, western blotting, and immunofluorescence assays.** DNA was extracted using the PureLink Genomic DNA Mini kit (Invitrogen) for genotyping. Schizonts at ~5% parasitaemia were lysed using 0.1% saponin/PBS (Sigma) and prepared for Western Blotting by standard methods. The presence of FLAG (F1804, Sigma) or GFP (Roche) was probed using primary antibody at 1:1000 dilution with the appropriate HRP-coupled secondary antibody added at 1:3000 dilution. Detection was performed with ECL (Amersham).

Schizonts at 5% parasitaemia or higher were treated with cysteine protease inhibitor E64 (10 μM) for 4–6 h to prevent egress, then fixed with 4% PFA/0.025% glutaraldehyde/PBS for 1 h, permeabilised with 0.1% Triton X-100/PBS for 10 min and blocked overnight at 4 °C in 3% BSA/PBS. Fixed cells were incubated with αFLAG (F1804, Sigma), αGAP45[58] and αGFP antibodies (Roche) at 1:500 dilution in 3% BSA/PBS for 2 h, then washed three times in PBS. Secondary antibodies were added at 1:1000 dilution in 3% BSA/PBS and incubated for 1 h, then cells were washed three times in PBS. The cells were resuspended to a hematocrit of 10% and mounted with DAPI-VECTASHIELD (Vectorlabs). Images were acquired with an OrcaFlash4.0 CMOS camera using a Nikon Ti Microscope (Nikon Plan Apo 100 × 1.4-N.A. oil immersion objective). Z-stacks were acquired with a step size of 0.2 μm and images were deconvoluted using the EpiDEMIC plugin with 50 iterations in Icy[59]. Subsequent image manipulations were carried out in ImageJ2[60–62].

**Conditional myosin knockout and quantification.** 30 ml of synchronous, late stage parasites at 5% parasitaemia were purified by density gradient centrifugation with 70% Percoll supplemented with sorbitol[63]. The purified schizonts were incubated with fresh RBCs at 1% hematocrit for 2–3 h, shaking at 100 rpm. Once new rings reached >10% parasitaemia the remaining schizonts were lysed by sorbitol treatment and rings were split equally into two 10 ml dishes at 2–4% parasitaemia, 4% hematocrit and incubated for 16 h with 0.05% DMSO or rapamycin (100 nM, Sigma), as well as heparin to prevent any further invasion (Pfizer, 1:25 dilution). Cultures were washed twice in complete media to remove heparin and DMSO/RAP, then transferred to 96 well plates in triplicate at 2% parasitaemia, 0.15% hematocrit for quantification, with invasion inhibitors heparin (1:25) or CytoD (500 nM) added to control samples. Parasites were allowed to proceed to the following cycle, and at 16–24 h post invasion were stained with SYBR Green I (Sigma) at 1:5000 dilution for 15 min. After three washes in PBS the parasites were analyzed by flow cytometry. Gates were set for RBCs, single cells and SYBR Green-positive cells (Supplementary Fig. 11).

**Reporting summary.** Further information on research design is available in the Nature Research Reporting Summary linked to this article.

## Data availability

All data are available from the corresponding author upon reasonable request. The crystallographic structures can be found on the Protein Data Bank (PDB codes 6I7D and 6I7E). The source data underlying Figs. 1e, 5b–e and Supplementary Fig 7 are provided as a Source Data file.

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

## Acknowledgements

We thank Margaret Titus for critical reading of the paper. We are grateful to beamline scientists of PX1 and PX2A (SOLEIL synchrotron) for excellent support during data collection. We thank Kathleen Ruppel and James Spudich (Stanford University) for the utrophin DNA. We thank Linda Makhlouf for experimental support. This work was funded by National Institutes of Health grant AI 132378 (A.H. and K.M.T.), and HL 124041 (M.J.P.). Parasite work was funded through an Investigator Award (100993/Z/13/Z, J.B.) and PhD studentship (109007/Z/15/A, T.B.) from Wellcome and through a program grant from the Human Frontier Science Program (RGY0066/2016, A.H. and J.B.). We thank Dr. Stéphane Réty for his advice and help to process the dataset of PfMyoA PPS.

## Author contributions

J.R.P., J.P.R., J.B., K.M.T., and A.H. designed the research. J.R.P., G.J., and D.M. crystallized PfMyoA in the different states. J.R.P. solved the structures and performed refinement. D.A. performed molecular dynamics. J.P.R. and C.S.B. performed in vitro functional assays. E.B.K. expressed and purified protein for crystallization. M.J.P. performed mass spectrometry analysis. T.B. generated conditional PfMyoA knockout. J.R.P., J.P.R., J.B., K.M.T., and A.H. discussed the results and wrote the paper with the help of D.A. and D.M.

## Additional information

**Competing interests:** The authors declare no competing interests.

