## [Peer Review File · Nature Communications]

Reviewers' Comments:

Reviewer #1:

Remarks to the Author:

In their manuscript Robert-Paganin et al. characterize the myosin motor MyoA of the highly medically relevant malaria parasite *Plasmodium falciparum*. Using conditional knock-out experiments they could show that PfMyoA is essential for the invasion of red blood cells through merozoites. Furthermore, the authors solved crystal structures of three states within the myosin motor cycle, namely the rigor-like, post-rigor and pre-powerstroke state. Based on these structures they could present a model of an atypical force generation, which relies on the N-terminal extension of PfMyoA including a phosphorylated SER instead of a fulcrum, a flexible residue allowing the piston-like movement of the SH1 helix in conventional myosins during their motor cycle. Additional functional characterization as well as in silico analysis of site directed mutants confirm the importance of the N-terminal extension, especially Ser19. Phosphorylation of this residue can tune force at the expense of speed, a mechanism hitherto not reported for other myosins and possibly tailored for the demands within the complex life cycle of *Plasmodium*.

Robert-Paganin et al.'s results are of great interest and therefore appropriate for publication in *Nature Communications*. As far as I can judge, their methodology is thorough and the data is of high quality. The presentation within this manuscript, however, needs major improvement to make their research accessible to non-expert readers and match the standards of this journal.

General points:

- The manuscript needs to be restructured thoroughly. It should have a clear abstract, introduction and results part. Now the introduction is partially moved to figure legends and mixed with results within the main text. Using headings within the results section would also significantly improve the reading.
- In their revised manuscript the authors should give a proper introduction about the structure and mechanism of canonical myosins for non-expert readers. Ex. what is the Relay, Wedge (residue range), what conformational changes occur within the motor cycle? To illustrate this, I suggest to edit Extended data figure 3 to focus on Myo2 only (3b should be moved to results figure to avoid confusion). The figure should furthermore be accompanied by a movie showing a morph (not two still images, they are less helpful than a figure) of Myo2 (ideally same view as used later for the PfMyoA morph).
- The manuscript needs a proper discussion with clear summaries and implications (rather than spreading it across the manuscript and legends).
- The scholarly presentation is also rather moderate. There are other related works available (partially from competing groups), that are not mentioned at all. Reworking the introduction and discussion part of the manuscript will give the chance to improve the scholarly presentation.
- Figure legends need to be shortened to stay informative. They are not meant to replace main text or repeat it. Necessary information should be moved to the main text; for example the conditional knock-out showing that MyoA is essential for merozoite invasion is treated in a single sentence in the main text only, despite being a key finding. The same is true for details of the crystal structures.
- References of figures need to be adjusted accordingly to changes (see previous point and detailed suggestions for figures below). Especially Extended Data Fig. 1 should not be cited whenever it is slightly related.
- The videos/ animations are not helpful the way they are now. The authors should prepare morphs of the states that help the reader to trace the structural elements. Additional labels/ arrows to highlight important regions and their movement would also be helpful to guide the reader's eye. As the mechanism is rather complex, I would suggest focusing on one aspect after the other to avoid

overloading the movie.

- The figures showing structural features are in general very crowded with many labels, highlights and colors. I recognize the difficulty to visualize a complex process like the myosin motor cycle and appreciate the usage of a consistent color code. A proper introduction of this code is however missing, so the reader might not benefit from it the way he/she could. The authors could furthermore improve the accessibility by including colors into their legends ex. Wedge (yellow), SH1 helix (red). Close-up views need arrows/angles to know the relative orientation relative to the full view. Furthermore, not all highlights and colors used in figures (ex. Pink balls, red dashed circles) are explained. Finally, using the color of the secondary structure for the residue labels might help the reader to orient. In the end, the authors might want to remove some of the highlights as they tend to be distracting if there are too many.
- The manuscript would benefit from a detailed and especially systematic comparison of scallop Myo2 with PfMyoA. The authors should clearly point out what is common and what not for each state (morphs/ superpositions will be helpful).
- The authors should moreover clarify that and why they are using crystal structures of Myo2 of two different organisms and justify why this is reasonable (sequence identify/ homology).
- Now that finally crystal structures of PfMyoA and PfAct1 as well as a cryo-EM structure of filamentous PfAct1 are available (Vahakoski et al. (2014), Pospich et al. (2017)), it's about time to put these together. Solving the complex structure of MyoA-Act1 is surely beyond the scope of this manuscript, a docking experiment (similar to what is shown in Fig. 1) could however already be of great interest. The authors should include a short paragraph about possible interactions and differences to conventional actin-myosin complexes in their revised manuscript.
- Based on the details provided by the MD simulations I have some problems following the authors' conclusions. First, was only one or were several simulations performed? Second, 60 ns is a rather short timeframe, why was it chosen? In Extended Data Fig.6 what is shown in a) and b)? The figure indicates it's the RMS deviation but the legend says RMS fluctuations. The RMSF using the initial structure (WT) as a reference would be of interest. Seeing 6c) I am wondering how this matches the described progressive displacement of the converter and Relay in the mutant shown in c). A video of the simulated frames would be very helpful to convince the reader of this conclusion.

Specific points:

Main text

- Line 38/Line 41 – move reference to Extended Data Fig. 1 to line 41, as its related to the life cycle and not the role of MyoA in the life cycle
- Line 40 - 'This contrasts with observations from *Toxoplasma gondii*, [...] suggesting PfMyoA is an essential motor [...].' – Wrong causality, not the contrast but the author's results suggest PfMyoA's essential role
- Line 44 – give residue number of SH1-SH2Gly (G695?)
- Lines 43 ff – Reference not clear as information about conventional myosin and PfMyoA are mixed. I suggest describing conventional myosins (Myo2) first and then continuing with differences in PfMyoA (see general points above)
- Line 57 f – PDB accession codes should be included in the main text to improve accessibility.
- Line 57 ff – The authors are not consistent about the usage of 'rigor' and 'rigor-like' state. As a crystal structure in absence of actin is clearly in a 'rigor-like state' (independent of how close this state is/might be to the real rigor state), the authors should be precise throughout the manuscript/ figure legends.
- Line 66 – 'Structural analysis shows a new compensatory mechanism' – replace new by hitherto unknown/ previously not reported
- Line 124 f – 'Because we show that PfMyoA is required for both motility and invasion [...]' –the authors only tested its role in the invasion process and referred to Siden-Kiamos et al. 2011 regarding

MyoA's role in motility

- Line 291 – Which actin was used? How was it expressed/ purified? Stabilized?
- Line 312 – Please include reference/ link to software package.
- Line 323 – Considering the low salt concentration of 5 mM it would be interesting if stabilized actin was used. Based on Zimmermann et al. (2015) details about the stabilizing agent and when it was added would should be included/discussed as they might affect the results.

Figures and legends

- Fig 1 - Color code needs explanation. The lever and light chains were not mentioned in the main text, are they resolved? Legend should include details/ why shown the way they are. Abbreviations like ELC should be introduced. Where does the actin structure come from? On what model is the docking based? The orientation of insert b) relative to the shown post-rigor structure should be clarified (magnified area, relative angle between views)
- Fig 2 – 'Upon binding to F-actin' is misleading, as the authors do not have a complex structure. What do the red dashed circles highlight? 2a) and 2c) almost show the same view, the authors might want to either combine those into one figure or choose a slightly different view for one of them/ highlight a different aspect.
- Fig 3 – Legend is missing the temperature used for experiment 3d). Change last reference to 'For data of higher utrophin concentrations see Extended Data Fig. 8'

Extended Data

- Fig. 1 – As the authors seem to focus on possible treatments of malaria within the life cycle the legend should be modified accordingly (see general points). Heading could for example read 'Plasmodium falciparum life cycle with focus on available treatment'. 'Schizont egress occurs every 48 hours', 'ready to be ejected on the next feed of on a new human host'. 'Current antimalarial therapeutics for malaria, targeting [...]'. Change to '[...] combination therapies based on artemisinin (ACTs)'. '[...] which likely work in many ways [...] – is the mode of action in general unknown? Add appropriate citation. 'Recent reports of resistant resistance to ACTS suggests suggest this frontline [...]'. Although no therapeutic, the authors should possibly mention vector control as a method of transmission blocking, considering its importance (see WHO World Malaria Report). The complete last paragraph should be shortened and incorporated into the main text 'The function of PfMyoA [...] (if carried into a blood meal).'
- Fig. 2 – I suggest moving this figure to the main article as it supports a key finding. '[..] (B9 and H6) or WT with RAP WT or with RAP [...]'. Missing citation(s) for invasion inhibitors heparin and CytoD (original research papers). Abbreviation RBCs not introduced previously. 'Act1 was used [...]' – specify organism of actin.
- Fig. 3 – See suggestions in general points, especially moving 3b to another figure to avoid confusion. Mentioning residues important for stability of the rigor conformation of PfMyoA before showing the structure on which this interpretation is based is incoherent. Specify legend heading. Properly label the Fulcrum (difficult to find in 3c). Add information about source organism in top panel of 3c. 3c) lower panel, position of lever arm should be indicated to orient more easily. 3D arrangement of helices close to Fulcrum is not visible in this figure, an additional video with morph should be included (see above). Abbreviations ELC and RLC not introduced/ mentioned in main text. 'ATP binding in myosin myosin's active site'. 'rearrangements during the powerstroke that couples couple'. '[..] a zoom on the Wedge show shows [...]'. 'Changes in the Switch-2 conformation pushes push [...]'
- Fig. 4 – A morph would improve the accessibility greatly (see general points). Extended Data Figure 3 and 4 show very similar views. The authors might want to modify figures to be more focused and thus precise. How are panels b) and c) related to the view in a)? Label for Y583 missing in b) although mentioned in legend. Why is the residue label I494 highlighted in pink? Arrows highlighting the movement of domains might be depicted in a different style/color as they do not stand out compared

to arrows used for residue labels. Additional arrows for the Switch-2 and SH1 helix might be helpful in b). Legend is missing explanation of pink sphere (G695). 'Switch-2 rearrangements displaces displace' 'As with PfMyoA (see Fig. 2b)' should be removed from Legend, as the reader does not have seen the structure of PfMyoA or Fig. 2 yet when Extended Data Fig. 4 is referred to.

- Fig. 5 – Reference of 5e) missing in legend and 5d) placed in wrong position.
- Fig. 6 – See general points above. Legend for 6d) is missing.
- Fig. 7 – '12% SDS-PAGE of PfMyoA [...] run on 12% SDS-gels', 'Y-axis was increased 40-fold' – X or Y axis? Legend and arrow in figure contradict each other.
- Fig. 8 – 'Expanded y-axis scales' and 'at higher utrophin concentrations shown' add compared to Figure 3
- Fig. 9 – Unfortunately, I cannot see stereo views myself despite trying hard. Assuming some reader might have the same problem, I suggest preparing an additional movie.
- Extended Data Movie 1 – see general points for all movies. 'In PfMyoA, [...] In PfMyoA', remove one occurrence.
- Extended Data Movie 2 – 'Conformational changes occurring within close to the [...]'
- Extended Data Movie 3 – 'on in the center'. Authors need to clarify how the T586 mutant structure was generated (as crystal structure was not solved). I am missing a final conclusion about the role of T586 (would also be of interest in the main text).

Reviewer #2:

Remarks to the Author:

Summary:

In this study the authors succeeded to determine the structure of the unconventional myosin MyoA of *Plasmodium falciparum*. This myosin is the core motor of the parasites glideosome that is required for efficient host cell invasion. Using a conditional gene excision system, the authors demonstrate that this protein is indeed essential for merozoite invasion, making it a good target for intervention strategies.

The authors then solved the structure of three states of the PfMyoA motor domain and visualised the conformational changes of this unconventional fast motor. Based on these results the authors hypothesised that the unusual large N-terminal extension of PfMyoA is required for efficient motor activity. To test this hypothesis, the authors tested PfMyoA versions with N-terminal deletions or point mutations in in vitro motility assays. Deletion of the N-terminus resulted in significantly lower ADP-release rates, resulting in 17-fold lower speeds, as measured by actin displacement. Similarly, the point mutation S19A resulted in lower speed, but higher force generation, suggesting that phosphorylation of S19 is required to tune MyoA motor properties.

Own opinion:

This is an excellent study and solves the conundrum regarding the motor mechanism of the unconventional myosin A, which is the core motor of the parasites glideosome. From what this reviewer can tell (not an expert in crystallography/structural biology), the study has been performed very rigorous, making use of several critical mutants to test the hypothesis that the N-terminal extension of MyoA is critical for motor function and tuning of its activity by phosphorylation of S19. The conclusions drawn by the authors are justified by the data.

Minor comments:

While this reviewer agrees that MyoA (and actin) of *Plasmodium* are essential for invasion, in contrast to *Toxoplasma gondii*, there is still an open debate if this is due to redundancies in the motor repertoire of *Toxoplasma gondii* or due to other factors, including activity of the host cell. A recent study by Bichet et al., 2017 demonstrated that *T.gondii* parasites lacking MyoA are capable to invade

the host cell due to host cell activity and conclude that there is likely no redundancy in the repertoire of myosins in *Toxoplasma*. Furthermore, similar to the situation with MyoA, analysis of conditional mutants for actin in *Plasmodium* and *Toxoplasma* demonstrated that PfAct1 is essential for invasion of *P.falciparum*, while parasites depleted for TgAct1 show residual invasion in case of *T.gondii*.

Therefore, this reviewer suggests to present a more balanced interpretation, when comparing the role of MyoA between *Plasmodium* and *Toxoplasma*.

Reviewer #3:

Remarks to the Author:

The authors solved crystal structures for the *Plasmodium falciparum* myosin A (PfMyoA) motor domain in three states, a rigor, a post-rigor and a pre-powerstroke state. They report that the N-terminal extension, specifically the phosphorylated serine 19, is critical for allosteric conformational changes for this myosin motor. Using solution kinetic studies and gliding filament assays they report that unphosphorylated PfMyoA generates more force and less speed in the gliding filament assay compared with phosphorylated PfMyoA. They conclude that Ser19 phosphorylation at the N-terminus of PfMyoA can tune the mechanical output of this motor. They propose a model according to which PfMyoA is phosphorylated in sporozoites, which enables the motor to move actin at high speed ($>2 \mu\text{m/s}$), while in the merozoite stage PfMyoA is unphosphorylated at Ser19 and tuned to generate a stronger ensemble force during the parasite invasion process, so that phosphorylation of the N-terminal extension of PfMyoA might act as a switch to tune motor activity depending on the stage of the parasite.

These are interesting observations and the work presented is of good quality. As far as I am aware, the proposed mode of regulating *Plasmodium* parasite motility is novel and will be very interesting to others in the community, however, several points have to be addressed; the work also needs to be discussed a bit more in the context of recently published data by other groups, who also reported in vitro motility assays and ATPase measurements for *Plasmodium* myosin A (e.g. Green et al JBC 2017).

Main points:

1. Unfortunately, the crystal structures of the motor domains do not seem to include the lever arm with the light chains; the lever arm plus light chains seems to have been added in an estimated orientation, which makes this critical part of the structure difficult to assess; line 60: “.. The amplitude of the subdomain movements and lever arm swing are similar to that of other myosins..”. Are there structural data on the lever arm and its swing between different nucleotide states available for PfMyoA and how do these structures compare to other myosins?
2. The authors estimate the amplitude of the myosin A power stroke from the modelled lever arm. It would be important to compare this estimate with the myosin A power stroke that has been measured in recent single molecule mechanical experiments using an optical trap (Green et al JBC 2017) and to discuss it in this context. Given the mechanical data, could the lever arm swing be smaller than that of other myosins for example?
3. The authors propose that phosphorylation of Ser 19 is serving as the switch to change the myosin

motor mechanics between the sporozoite and the merozoite stage. Line 95: Ser19 phosphorylation occurs in vivo and tunes the PfMyoA properties. This is important for assessing the model (Fig. 4), and the authors should expand on this point; phosphorylation state of myosin A in sporozoites compared to merozoites? How does myosin A phosphorylation/dephosphorylation in sporozoites and merozoites affect the gliding motility and invasion of the parasite? Potential regulatory role of other components such as the GAPs?

4. The difference in speed in the in vitro motility in the absence of phosphorylation and N-terminus is very interesting and should be compared with other modes of PfMyoA motor regulation, such as the light chains. The actin-activated ATPase experiments on myosin A should also be compared with the actin-activated ATPase data reported recently by other groups. What is the stoichiometry of the light chains MTIP and PfELC in the motility assays?

Detailed Response to Reviewers:

Paganin et al. 2019

Plasmodium myosin A drives parasite invasion by an atypical force generating mechanism

To the Reviewers:

We have rewritten the manuscript text to conform to the format required for a Nature Communications article, as the reviewed manuscript was written in Nature format. The changes are thus extensive and could not be shown with track changes. The new manuscript describes in depth the new findings as suggested by Reviewer 1. In addition, several elements of discussion were also added in this new expanded version.

The main changes in the figures in this new version are highlighted below:

Previous version	New version	Figures
Supplementary 2	Figure 1	PfMyoA is essential for red blood cell invasion by merozoites
Figure 1	Figure 2	Structural states and motor cycle of PfMyoA
Supplementary 3b	Figure 3	Sequence alignment of connectors essential in driving motor conformational changes – lack of canonical residues in PfMyoA
Figure 2	Figure 4	The unconventional mechanism of force production by PfMyoA
Figure 3	Figure 5	Functional properties of wild-type (WT) and mutant full-length PfMyoA constructs.
Figure 4	Figure 6	Phosphorylation of PfMyoA tunes its motor properties

Supplementary Figures

Supplementary 3a	sup Fig 2	
Supplementary 5	sup Fig 3	
Supplementary 3c	sup Fig 4	
Addition of	sup Fig 10	comparison of the Nter extension location in Myo1b and PfMyoA
Addition of several movies as requested by reviewer 1 to compare Myo2 and PfMyoA and to illustrate the results of molecular dynamics.		

Answer to reviewers are added below their comments in the following:

Reviewer #1 (Remarks to the Author):

In their manuscript Robert-Paganin et al. characterize the myosin motor MyoA of the highly medically relevant malaria parasite *Plasmodium falciparum*. Using conditional knock-out experiments they could show that PfMyoA is essential for the invasion of red blood cells through merozoites. Furthermore, the authors solved crystal structures of three states within the myosin motor cycle, namely the rigor-like, post-rigor and pre-powerstroke state. Based on these structures they could present a model of an atypical force generation, which relies on the N-terminal extension of PfMyoA including a phosphorylated SER instead of a fulcrum, a flexible residue allowing the piston-like movement of the SH1 helix in conventional myosins during their motor cycle. Additional functional characterization as well as in silico analysis of site directed mutants confirm the importance of the N-terminal extension, especially Ser19. Phosphorylation of this residue can tune force at the expense of speed, a mechanism hitherto not reported for other myosins and possibly tailored for the demands within the complex life cycle of *Plasmodium*.

Robert-Paganin et al.'s results are of great interest and therefore appropriate for publication in Nature Communications. As far as I can judge, their methodology is thorough and the data is of high quality. The presentation within this manuscript, however, needs major improvement to make their research accessible

to non-expert readers and match the standards of this journal.

We thank Reviewer #1 for the positive appreciation of our findings and for his useful advices and critique. Initially, this work was written in a letter format for *Nature*, which is the reason it was not sent to reviewers in the format usually found for *Nature communications* articles. We have now rewritten the article taking into account the comments on style provided by Reviewer 1. All the points he requested were addressed; in particular we made sure to provide the background requested to make the findings easily accessible to readers of a general audience.

General points:

- The manuscript needs to be restructured thoroughly. It should have a clear abstract, introduction and results part. Now the introduction is partially moved to figure legends and mixed with results within the main text. Using headings within the results section would also significantly improve the reading.

We wrote a new version that takes into account this remark. All the different headings are present and Results and Discussion sections are clearly distinguished.

- In their revised manuscript the authors should give a proper introduction about the structure and mechanism of canonical myosins for non-expert readers. Ex. what is the Relay, Wedge (residue range), what conformational changes occur within the motor cycle? To illustrate this, I suggest to edit Extended data figure 3 to focus on Myo2 only (3b should be moved to results figure to avoid confusion). The figure should furthermore be accompanied by a movie showing a morph (not two still images, they are less helpful than a figure) of Myo2 (ideally same view as used later for the PfMyoA morph).

In the new version, basic description of the myosin motor domain has been added in the introduction (**lines 88-92 and Supplementary Fig.2**).

In the section '**Results**', a paragraph has been added to introduce the mechanism of canonical myosins ('**Rearrangements in connectors of classical myosins**', **lines 166-190**), it is followed by a paragraph comparing it with PfMyoA ('**Rearrangements in connectors of PfMyoA define an atypical structural motor mechanism**', **lines 192-216**). The figures have been changed to illustrate the changes in the Relay/Wedge/SH1 helix found in connectors and the previous **Extended data Figure 3b** is now moved to **Figure 3** as suggested by the reviewer.

The notion of connectors rearrangement driving the motor conformational changes has been introduced in the text ('**Rearrangements in connectors of classical myosins**', **lines 166-190**) and in the figure legend of **Supplementary Fig. 2**. We do not think morphing movies are appropriate for the powerstroke, since they provide a wrong information considering that the powerstroke happens in several transitions currently unknown. We changed however the timing between the images to better visualize the animation and added two other animations (**Supplementary Movies 1, 2, 3, 4**).

- The manuscript needs a proper discussion with clear summaries and implications (rather than spreading it across the manuscript and legends).

A **Discussion** section (**lines 264-366**) has been developed in the new version of the manuscript. In particular the comparison with the role of the N-terminal extension of Myo1b in defining the duty-ratio has been expanded.

- The scholarly presentation is also rather moderate. There are other related works available (partially from competing groups), that are not mentioned at all. Reworking the introduction and discussion part of the manuscript will give the chance to improve the scholarly presentation.

This has been corrected in the new version of the manuscript both in introduction, results and in discussion sections. A general description of what is known for myosin structural rearrangements is presented as well as the description of the work recently published on the structure of the *Toxoplasma* MyoA Pre-powerstroke state (Powell *et al.*, 2018) see **lines 95-99, 269-273, 324-327**.

- Figure legends need to be shortened to stay informative. They are not meant to replace main text or repeat it. Necessary information should be moved to the main text; for example the conditional knock-out showing that MyoA is essential for merozoite invasion is treated in a single sentence in the main text only, despite being a key finding. The same is true for details of the crystal structures.

This has been corrected and the text now is detailed to describe the results while the figure legends are minimized in length (< than 350 words, as expected in *Nature Comms* format).

- References of figures need to be adjusted accordingly to changes (see previous point and detailed suggestions for figures below). Especially Extended Data Fig. 1 should not be cited whenever it is slightly related.

We modified the Figures, as suggested by this reviewer (see the points below).

- The videos/ animations are not helpful the way they are now. The authors should prepare morphs of the states that help the reader to trace the structural elements. Additional labels/ arrows to highlight important regions and their movement would also be helpful to guide the reader's eye. As the mechanism is rather complex, I would suggest focusing on one aspect after the other to avoid overloading the movie.

As stated earlier, we do not think that Morphs are appropriate to describe the powerstroke since this is a multi-step event. We have however simplified the movies and multiplied them to help the reader. Having the PfMyoA/Myo2 images below one another rather than one next to each other should also help the reader to appreciate the common and divergent aspects of the motor mechanism in both systems. In order to clarify this point, we added a few lines in Methods "**Motor cycle reconstitution and visualization**" (lines 556-570) with a paragraph specifically addressing how to best compare the structures with animations: "In this study, we reconstitute the motor cycle based on three structural states: post-rigor (PR), pre-powerstroke (PPS) and rigor-like. To visualize structural changes during the powerstroke, we used a comparison between the PPS and the rigor-like states. We have made some animations (**Supplementary Movies 1-4**) to compare the conformation of the different connectors in the PPS and in the Rigor-like states in conventional myosins (ScMyo2) and in the atypical PfMyoA. Morphs were not used to illustrate the transition between the PPS and the rigor-like since the isomerization occurs through intermediate states⁵² which are not known for these myosins, this kind of procedure would thus be misleading."

- The figures showing structural features are in general very crowded with many labels, highlights and colors. I recognize the difficulty to visualize a complex process like the myosin motor cycle and appreciate the usage of a consistent color code. A proper introduction of this code is however missing, so the reader might not benefit from it the way he/she could. The authors could furthermore improve the accessibility by including colors into their legends ex. Wedge (yellow), SH1 helix (red). Close-up views need arrows/angles to know the relative orientation relative to the full view. Furthermore, not all highlights and colors used in figures (ex. Pink balls, red dashed circles) are explained. Finally, using the color of the secondary structure for the residue labels might help the reader to orient. In the end, the authors might want to remove some of the highlights as they tend to be distracting if there are too many.

We carefully modified the figures to make them clearer and less confusing regarding the labels. Some labels were removed in order to make them less crowded while some side-chains were coloured to visualize them better. The figures and the movies are all consistent with the same colour code throughout. The color code has been properly introduced in the legend of **Fig.2, Fig.3, Fig. 4, Supplementary Fig. 2, Supplementary Fig. 5**. In the new legends, we introduced and defined all the elements of the figures as well as their colour.

- The manuscript would benefit from a detailed and especially systematic comparison of scallop Myo2 with PfMyoA. The authors should clearly point out what is common and what not for each state (morphs/ superpositions will be helpful).

Superimpositions are not the best way to illustrate the differences but we have made movies that alternate rapidly between the two myosins to see more clearly the differences (**Supplementary Movies 1-4**). In particular, **Supplementary Movie 1** alternate between PfMyoA and ScMyo2 corresponding to the same structural state. We also consequently modified the text in order to make the comparison clearer

(‘Rearrangements in connectors of classical myosins’, lines 162-186, ‘Rearrangements in connectors of PfMyoA define an atypical structural motor mechanism’, lines 188-212).

- The authors should moreover clarify that and why they are using crystal structures of Myo2 of two different organisms and justify why this is reasonable (sequence identify/ homology).

The structures of the PPS and the Rigor-like states of ScMyo2 have been solved from two different organisms, but the sequences of the heavy chain of these myosins are highly similar. The two structures can thus be used to illustrate the powerstroke in ScMyo2. We wrote a paragraph in the section **“Motor cycle reconstitution and visualization” (Lines 556-570)** of the Methods to explain this point: “In order to visualize the powerstroke of a conventional class II myosin, we have used scallop myosin II (ScMyo2). The structures of the Pre-powerstroke (PPS) and of the Rigor-like states of ScMyo2 have been solved in different organisms: PPS of bay scallop (*Argopecten irradians*) ScMyo2 (PDB code 1QVI, resolution 2.54 Å) and Rigor-like state of Atlantic deep-sea scallop (*Placopecten magellanicus*) ScMyo2 (PDB code 2OS8, resolution 3.27 Å). The sequence of ScMyo2 heavy chain in these two species is highly conserved for the motor domain (92% identity, 95% similarity) and the two structures can thus be used to reconstitute the powerstroke of ScMyo2.”

- Now that finally crystal structures of PfMyoA and PfAct1 as well as a cryo-EM structure of filamentous PfAct1 are available (Vahakoski et al. (2014), Pospich et al. (2017)), it’s about time to put these together. Solving the complex structure of MyoA-Act1 is surely beyond the scope of this manuscript, a docking experiment (similar to what is shown in Fig. 1) could however already be of great interest. The authors should include a short paragraph about possible interactions and differences to conventional actin-myosin complexes in their revised manuscript.

Actomyosin structures are difficult to predict. The currently high-resolution structures of actomyosin determined by CryoEM differ from one another in a non-predictable way. The interface and motor domain/F-actin orientation can differ (Von der Ecken *et al.*, 2016; Guren *et al.*, 2017; Menten *et al.*, 2018). It is not possible to predict how PfAct1 and PfMyoA interact and we decided not to discuss this point since our structures do not include actin. The mechanism described in this paper does not involve directly the actin binding interface. Previous data comparing Rigor-like structures and Actomyosin structures indicate that they do not differ greatly allowing us to focus our discussion on the unconventional mechanism of PfMyoA for force generation. Future cryoEM investigations will hopefully provide data to visualize the actomyosin interface of PfMyoA and the PfAct1 filament in detail.

- Based on the details provided by the MD simulations I have some problems following the authors’ conclusions. First, was only one or were several simulations performed? Second, 60 ns is a rather short timeframe, why was it chosen? In Extended Data Fig.6 what is shown in a) and b)? The figure indicates it’s the RMS deviation but the legend says RMS fluctuations. The RMSF using the initial structure (WT) as a reference would be of interest. Seeing 6c) I am wondering how this matches the described progressive displacement of the converter and Relay in the mutant shown in c). A video of the simulated frames would be very helpful to convince the reader of this conclusion.

The results of one simulation is presented here, but the routine used is robust and reproducible, and 60 ns is sufficient to appreciate the fluctuations in the mutant and compare it to the WT. For further detail about the procedure, see Robert-Paganin *et al.*, 2018. Corrections have been made on what is now **Supp. Fig. 8**. We have indeed corrected a mistake that was introduced in the previous version, RMSF are indeed calculated after fitting on the backbone atoms of the first frame (which are obviously equivalent in both cases since the mutant is virtually built on the solved structure of PfmyoA).

As suggested by the reviewer, two movies have been added to illustrate the simulated trajectories of the WT and the mutant (**Supplementary Movies 5-6**). We have also added a more detailed description of these results in the main text as follow (**Lines 252-262**). ‘The role of SEP19 in this transition was further investigated *in silico* using the mutant K764E. Molecular dynamic simulations on a 60 ns time-course further confirm the role of this electrostatic bond for the stability of the converter position, by comparing phosphorylated WT and K764E mutant in the Rigor-like state (**Supplementary Fig. 8**). In the WT phosphorylated motor domain, the converter is maintained in its Rigor-like position throughout the simulation (**Supplementary Fig. 8a, 8c, 8d, Supplementary Movie 5**). Conversely, in the K764E mutant, the

converter position is not maintained and progressively deviates from its initial position together with the Relay, while the SH1-helix stays immobile (**Supplementary Fig. 8b, 8c, 8d, Supplementary Movie 6**). The in silico and in vitro results confirm that the electrostatic interaction between SEP19 and K764 stabilizes the Rigor-like state and is important for modulating ensemble force and the speed at which PfMyoA moves actin.'

Specific points:

Main text

- Line 38/Line 41 – move reference to Extended Data Fig. 1 to line 41, as its related to the life cycle and not the role of MyoA in the life cycle

The life cycle is better described in introduction with a reference to **Supp. Fig. 1**.

- Line 40 - 'This contrasts with observations from *Toxoplasma gondii*, [...] suggesting PfMyoA is an essential motor [...].' – Wrong causality, not the contrast but the author's results suggest PfMyoA's essential role

The text has been changed in introduction :

'Several studies have investigated the role of myosin A in both the coccidian parasite *Toxoplasma gondii* and across the genus *Plasmodium*. In *Toxoplasma gondii*, Myosin A (TgMyoA) has been implicated in both invasion and egress of the parasite from the infected cell^{6,7}. TgMyoA is not, however, essential for invasion and its function can be compensated for by TgMyoB or its splicing isoform TgMyoC^{8,9}, or by active host cell-mediated internalization¹⁰. Although host cell membrane wrapping forces likely play a role in merozoite invasion of red blood cells¹¹, their role has not been tested in the absence of a parasite motor. In the *Plasmodium* genus, initial experiments using the mouse malaria parasite *P. berghei*, showed that PbMyoA is required for cell motility and midgut colonization during parasite mosquito stages¹². More recently, genetic ablation of GAP45 in *P. falciparum* blocked parasite invasion in merozoite stages, but not egress¹³. Whilst this study points to a critical role for the motor complex in blood stages of infection, the structural role GAP45 plays in apicomplexan cell architecture as well as glideosome function leaves the essential functionality of MyoA in the merozoite unresolved.' (**Lines 65-76**).

- Line 44 – give residue number of SH1-SH2Gly (G695?)

The residue number has been given: '(G695 in scallop myosin 2)' (**Line 87**).

- Lines 43 ff – Reference not clear as information about conventional myosin and PfMyoA are mixed. I suggest describing conventional myosins (Myo2) first and then continuing with differences in PfMyoA (see general points above)

As stated above, this has been corrected. The mechanism of classical myosins is first introduced prior to that of PfMyoA. Changes in sequence related to the lack of conservation of canonical residues are introduced upon this comparison.

- Line 57 f – PDB accession codes should be included in the main text to improve accessibility.

PDB accession codes have been included in the **Results**, in the first paragraph of the section: '**X-ray structures of three structural states of PfMyoA**' (**Line 139**) and they are also mentioned in the **Supplementary Tables 1-2**.

- Line 57 ff – The authors are not consistent about the usage of 'rigor' and 'rigor-like' state. As a crystal structure in absence of actin is clearly in a 'rigor-like state' (independent of how close this state is/might be to the real rigor state), the authors should be precise throughout the manuscript/ figure legends.

We have carefully checked and systematically refer to Rigor-like state in the manuscript whenever we are mentioning a crystal structure. Note however that it was established that Rigor-like and Rigor structures are similar with almost no change except on the actin-binding loops for myosins in which the 50 kDa cleft is fully closed in the Rigor-like state (i.e. Myo1b, NM2c). Note that this is the case for the PfMyoA Rigor-like structure. We have added the sentence: 'The PfMyoA rigor-like state adopts a fully closed cleft as found for

non muscle Myo2c and Myo1b Rigor states and is thus likely a good model of this myosin in the Rigor state' (Lines 147-148) in the text to describe this point.

- Line 66 – ‘Structural analysis shows a new compensatory mechanism’ – replace new by hitherto unknown/ previously not reported

This has been corrected. The sentence has been replaced by: ‘Structural analysis shows an unforeseen compensatory mechanism’ (Line 162).

- Line 124 f – ‘Because we show that PfMyoA is required for both motility and invasion [...]’ – the authors only tested its role in the invasion process and referred to Siden-Kiamos et al. 2011 regarding MyoA’s role in motility

This has been corrected. The sentence has been replaced by: ‘Given the combined evidence that MyoA is critical for both invasion and for motility (Siden-Kiamos *et al.*, 2011) in *Plasmodium* parasites, it would appear to be a bona fide first-order therapeutic target for preventing malaria parasite infection and disease progression.’ (Line 134-137).

- Line 291 – Which actin was used? How was it expressed/ purified? Stabilized?

Skeletal actin was purified from tissue. A reference to the method of purification is now included in **Methods** (last sentence of first paragraph of “**Protein expression and purification**”, line 497). We are more explicit in the legends and methods that skeletal actin was used for these studies. For actin-activated ATPase, there is no need to stabilize the actin, it remains polymerized. For *in vitro* motility, the skeletal actin was labeled with rhodamine-phalloidin, the standard method for visualizing actin when using this technique.

- Line 312 – Please include reference/ link to software package.

The requested information was added in **Methods** to the section entitled (“**Unloaded and loaded *in vitro* motility measurements**”, lines 572).

- Line 323 – Considering the low salt concentration of 5 mM it would be interesting if stabilized actin was used. Based on Zimmermann et al. (2015) details about the stabilizing agent and when it was added would should be included/discussed as they might affect the results.

For actin-activated ATPase activity measurements, the actin is not stabilized (see answer to line 291 query) so this is not a concern. The reference the reviewer mentions found some differences in processive run lengths for a class 5 and a class 6 myosin on “old” ADP actin versus “young” ATP/ADP.Pi actin (<300s old). By this nomenclature all the actin we used would be “old” actin that is at steady state.

The skeletal actin used for *in vitro* motility has phalloidin-rhodamine for visualization. In our prior publication (Bookwalter et al., JBC 2017:19290) we showed that phalloidin versus jasplakinolide stabilized skeletal actin was moved at the same speed by PfMyoA, and that jasplakinolide stabilized skeletal actin versus jasplakinolide stabilized *Plasmodium* actin were also moved at the same speed by PfMyoA.

Figures and legends

- Fig 1 - Color code needs explanation. The lever and light chains were not mentioned in the main text, are they resolved? Legend should include details/ why shown the way they are. Abbreviations like ELC should be introduced. Where does the actin structure come from? On what model is the docking based? The orientation of insert b) relative to the shown post-rigor structure should be clarified (magnified area, relative angle between views)

The ELC and MTIP are now defined in **Supp. Fig. 1** and **Figure 2**. The lever arm is not part of the X-ray structures, which contains only the motor domain (1-768), we added a schematic lever arm to help the reader to appreciate the lever arm swing. We added a sentence in the legend to make it clear: ‘To appreciate the movement of the converter and how it can be amplified by the rest of the lever arm, the IQ region and the two LCs (ELC and MTIP, Supplementary Fig. 1) are represented schematically in continuity of the last helix of the converter.’ (Lines 396-398).

The actin structure was initially taken from Stefan Raunser's structure (Von der Ecken *et al.*, 2016), but indeed it could be misleading. To avoid confusion, we replaced it by a scheme of the actin filament as it was done for previous Figure 4 (now Figure 6).

- Fig 2 – 'Upon binding to F-actin' is misleading, as the authors do not have a complex structure. What do the red dashed circles highlight? 2a) and 2c) almost show the same view, the authors might want to either combine those into one figure or choose a slightly different view for one of them/ highlight a different aspect.

To avoid confusion, the sentence is now: 'The Rigor-like structure indicates that the sequential release of hydrolysis products upon the powerstroke triggers displacement of the Wedge which is associated with straightening of the Relay helix and the converter swing.' (Figure 4 : Lines 428-430).

The legend of this figure (Figure 4) now defines what is highlighted with the dashed circles. Regarding figure 2a) and 2c) (which are now Figures 4a and 4c), we chose to show the same view for clarity for the reader, but they show each different types of interactions. Each image is connected to a different idea, 2a) is a general view of all the elements/connectors in the two states (PPS and Rigor-like) while 2c) focuses on the converter swing and on additional bonds stabilizing the Rigor-like state (electrostatic bond between the N-term extension and the converter; salt bridge-pi stacking between the N-term extension, Switch-1 and Switch-2). We prefer keeping the same orientation for the two images. But all this information cannot be visualized with only one figure.

- Fig 3 – Legend is missing the temperature used for experiment 3d). Change last reference to 'For data of higher utrophin concentrations see Extended Data Fig. 8'

The revised figure legend reads "Supp. Fig. 7b shows these force data and fits extended to higher utrophin concentrations. Supp. Fig. 7c-e shows ΔN data shown with an expanded y-axis. Skeletal actin was used for all experiments. Temperature, 30°C." Lines 465-467.

Extended Data

- Fig. 1 – As the authors seem to focus on possible treatments of malaria within the life cycle the legend should be modified accordingly (see general points). Heading could for example read 'Plasmodium falciparum life cycle with focus on available treatment'. 'Schizont egress occurs each every 48 hours', 'ready to be ejected on the next feed of on a new human host'. 'Current antimalarial therapeutics for malaria, targeting [...]'. Change to '[...] combination therapies based on artemisinin (ACTs)'. '[...] which likely work in many ways [...]' – is the mode of action in general unknown? Add appropriate citation. 'Recent reports of resistant resistance to ACTS suggests suggest this frontline [...]'. Although no therapeutic, the authors should possibly mention vector control as a method of transmission blocking, considering its importance (see WHO World Malaria Report).

The complete last paragraph should be shortened and incorporated into the main text 'The function of PfMyoA [...] (if carried into a blood meal).'

This figure (Supp. Fig. 1) has been substantially changed in the new version.

- Fig. 2 – I suggest moving this figure to the main article as it supports a key finding. '[..] (B9 and H6) or WT with RAP WT or with RAP [...]'. Missing citation(s) for invasion inhibitors heparin and CytoD (original research papers). Abbreviation RBCs not introduced previously. 'Act1 was used [...]' – specify organism of actin.

As suggested, we have moved this figure to the main article – now Fig. 1. Another study was for the the invasion inhibitors (Zuccala *et al.*, 2016). "RBCs" has now been defined (Lines 126, 375) and Act1 was changed to PfAct1.

- Fig. 3 – See suggestions in general points, especially moving 3b to another figure to avoid confusion. Mentioning residues important for stability of the rigor conformation of PfMyoA before showing the structure on which this interpretation is based is incoherent. Specify legend heading. Properly label the

Fulcrum (difficult to find in 3c). Add information about source organism in top panel of 3c. 3c) lower panel, position of lever arm should be indicated to orient more easily. 3D arrangement of helices close to Fulcrum is not visible in this figure, an additional video with morph should be included (see above).

Previously named Figure 3a and another figure of the motor domain were grouped to make **Supp. Fig. 3** in order to define the different parts of the myosin motor.

The sequence alignment (3b) has been moved to become **Figure 3**; structural features of Myo2 are clearly labeled to define the classical rearrangements of the Relay and SH1 helix. We have moved 3c to another figure (**Supp. Fig. 4**). The fulcrum is now precisely labeled.

Abbreviations ELC and RLC not introduced/ mentioned in main text. 'ATP binding in myosin myosin's active site'. 'rearrangements during the powerstroke that couples couple'. '[..] a zoom on the Wedge show shows [...]'. 'Changes in the Switch-2 conformation pushes push [...]'

The definition of these abbreviations has been introduced in the legend of **Supp. Fig. 2** and we mentioned the light chains in the main text (introduction): 'The N-terminal region of one of the light chains (LC), called myosin tail interacting protein (MTIP), is believed to anchor the myosin to a membrane bound complex of glideosome associated proteins (GAP45-GAP50-GAP40) (**Supplementary Fig. 1b**)' (**Lines 63-65**).

Supp Fig.2 figure legend : 'PfMyoA contains the essential light chain (ELC) and the myosin tail domain interacting protein (MTIP).'

- Fig. 4 – A morph would improve the accessibility greatly (see general points). Extended Data Figure 3 and 4 show very similar views. The authors might want to modify figures to be more focused and thus precise. How are panels b) and c) related to the view in a)? Label for Y583 missing in b) although mentioned in legend. Why is the residue label I494 highlighted in pink? Arrows highlighting the movement of domains might be depicted in a different style/color as they do not stand out compared to arrows used for residue labels. Additional arrows for the Switch-2 and SH1 helix might be helpful in b). Legend is missing explanation of pink sphere (G695). 'Switch-2 rearrangements displaces displace' 'As with PfMyoA (see Fig. 2b)' should be removed from Legend, as the reader does not have seen the structure of PfMyoA or Fig. 2 yet when Extended Data Fig. 4 is referred to.

As mentioned earlier, the morph is not a good idea to illustrate the changes that occur during the powerstroke but we changed the timing between the two states in the movies and we simplified the movies to help the readers to appreciate the changes.

- Fig. 5 – Reference of 5e) missing in legend and 5d) placed in wrong position.

These figures are now described in the legend (**Sup. Fig. 3e**) and in discussion (**lines 335-248**).

- Fig. 6 – See general points above. Legend for 6d) is missing.

This has been corrected. The figure is now **Sup. Fig. 8**.

- Fig. 7 – '12% SDS-PAGE of PfMyoA [...] run on 12% SDS-gels', 'Y-axis was increased 40-fold' – X or Y axis? Legend and arrow in figure contradict each other.

The first sentence of this legend has been revised. The confusing arrow was removed, and the text states that the y-axis was expanded 40-fold.

- Fig. 8 – 'Expanded y-axis scales' and 'at higher utrophin concentrations shown' add compared to Figure 3 The current Supplementary Figure 7 legend was changed as requested.

- Fig. 9 – Unfortunately, I cannot see stereo views myself despite trying hard. Assuming some reader might have the same problem, I suggest preparing an additional movie.

This stereo figure is a requirement of the *Nature Communications* format.

- Extended Data Movie 1 – see general points for all movies. 'In PfMyoA, [...] In PfMyoA', remove one occurrence.

This change was introduced in the legend of the movie which is now **Supplementary Movie 2**.

- Extended Data Movie 2 – ‘Conformational changes occurring within close to the [...]’

This change was introduced in the legend of the movie which is now **Supplementary Movie 3**.

- Extended Data Movie 3 – ‘on in the center’. Authors need to clarify how the T586 mutant structure was generated (as crystal structure was not solved). I am missing a final conclusion about the role of T586 (would also be of interest in the main text).

The legend of this Movie, now **Supplementary Movie 5** has been changed according to the reviewer’s comment. PfMyoA has a Threonine in its Wedge sequence rather than a bulky aromatic residue at this position of the wedge in canonical myosin sequences. The Tyrosine side chain was introduced in coot and the conformation found for the side chain in the PPS and Rigor-like structures of scallop were chosen. This allows here to illustrate how a bulky side chain would cause steric hindrance during the powerstroke since the wedge becomes closer to the fulcrum and this fulcrum is not bulkier since the canonical glycine ^{SH2}-^{SH1}Gly is in fact a serine in the PfMyoA sequence. The main text illustrates this point:

‘First, the aromatic and bulky residue of the Wedge is absent in PfMyoA and replaced by a threonine (T586), and the nearby bulky methionine residue from the Relay (^{Relay}511-DFGMD-515 in ScMyo2) is also absent and replaced by a threonine (T522) in PfMyoA (^{Relay}520-KYTS-523) (compare **Fig. 4b and Supplementary Fig. 5b; Supplementary Movie 4**). The movement of the Wedge during the powerstroke results in less steric hindrance and is thus adapted to the lack of mobility of the SH1-helix which stays in position while the Relay kink resolves and the lever arm swings (**Fig. 4b, 4c, Supplementary Movie 3**).’ (Lines 199-205).

Reviewer #2 (Remarks to the Author):

Summary:

In this study the authors succeeded to determine the structure of the unconventional myosin MyoA of Plasmodium falciparum. This myosin is the core motor of the parasites glideosome that is required for efficient host cell invasion. Using a conditional gene excision system, the authors demonstrate that this protein is indeed essential for merozoite invasion, making it a good target for intervention strategies. The authors then solved the structure of three states of the PfMyoA motor domain and visualised the conformational changes of this unconventional fast motor. Based on these results the authors hypothesised that the unusual large N-terminal extension of PfMyoA is required for efficient motor activity. To test this hypothesis, the authors tested PfMyoA versions with N-terminal deletions or point mutations in in vitro motility assays. Deletion of the N-terminus resulted in significantly lower ADP-release rates, resulting in 17-fold lower speeds, as measured by actin displacement. Similarly, the point mutation S19A resulted in lower speed, but higher force generation, suggesting that phosphorylation of S19 is required to tune MyoA motor properties.

Own opinion:

This is an excellent study and solves the conundrum regarding the motor mechanism of the unconventional myosin A, which is the core motor of the parasites glideosome. From what this reviewer can tell (not an expert in crystallography/structural biology), the study has been performed very rigorous, making use of several critical mutants to test the hypothesis that the N-terminal extension of MyoA is critical for motor function and tuning of its activity by phosphorylation of S19. The conclusions drawn by the authors are justified by the data.

We thank the reviewer for his positive assessment of our data and results.

Minor comments:

While this reviewer agrees that MyoA (and actin) of Plasmodium are essential for invasion, in contrast to Toxoplasma gondii, there is still an open debate if this is due to redundancies in the motor repertoire of Toxoplasma gondii or due to other factors, including activity of the host cell. A recent study by Bichet et al.,

2017 demonstrated that *T.gondii* parasites lacking MyoA are capable to invade the host cell due to host cell activity and conclude that there is likely no redundancy in the repertoire of myosins in *Toxoplasma*. Furthermore, similar to the situation with MyoA, analysis of conditional mutants for actin in *Plasmodium* and *Toxoplasma* demonstrated that PfAct1 is essential for invasion of *P.falciparum*, while parasites depleted for TgAct1 show residual invasion in case of *T.gondii*.

Therefore, this reviewer suggests to present a more balanced interpretation, when comparing the role of MyoA between *Plasmodium* and *Toxoplasma*.

We have expanded on the causes of residual invasion in *T. gondii* in the introduction, adding a reference to Bichet et al, 2017, and compared this to the situation in *P. falciparum* (see introduction, 2nd paragraph, lines 65-76). 'Several studies have investigated the role of myosin A in both the coccidian parasite *Toxoplasma gondii* and across the genus *Plasmodium*. In *Toxoplasma gondii*, Myosin A (TgMyoA) has been implicated in both invasion and egress of the parasite from the infected cell^{6,7}. TgMyoA is not, however, essential for invasion and its function can be compensated for by TgMyoB or its splicing isoform TgMyoC^{8,9}, or by active host cell-mediated internalization¹⁰. Although host cell membrane wrapping forces likely play a role in merozoite invasion of red blood cells¹¹, their role has not been tested in the absence of a parasite motor. In the *Plasmodium* genus, initial experiments using the mouse malaria parasite *P. berghei*, showed that PbMyoA is required for cell motility and midgut colonization during parasite mosquito stages¹². More recently, genetic ablation of GAP45 in *P. falciparum* blocked parasite invasion in merozoite stages, but not egress¹³. Whilst this study points to a critical role for the motor complex in blood stages of infection, the structural role GAP45 plays in apicomplexan cell architecture as well as glideosome function leaves the essential functionality of MyoA in the merozoite unresolved.'

Reviewer #3 (Remarks to the Author):

The authors solved crystal structures for the *Plasmodium falciparum* myosin A (PfMyoA) motor domain in three states, a rigor, a post-rigor and a pre-powerstroke state. They report that the N-terminal extension, specifically the phosphorylated serine 19, is critical for allosteric conformational changes for this myosin motor. Using solution kinetic studies and gliding filament assays they report that unphosphorylated PfMyoA generates more force and less speed in the gliding filament assay compared with phosphorylated PfMyoA. They conclude that Ser19 phosphorylation at the N-terminus of PfMyoA can tune the mechanical output of this motor. They propose a model according to which PfMyoA is phosphorylated in sporozoites, which enables the motor to move actin at high speed (>2 $\mu\text{m/s}$), while in the merozoite stage PfMyoA is unphosphorylated at Ser19 and tuned to generate a stronger ensemble force during the parasite invasion process, so that phosphorylation of the N-terminal extension of PfMyoA might act as a switch to tune motor activity depending on the stage of the parasite.

These are interesting observations and the work presented is of good quality. As far as I am aware, the proposed mode of regulating *Plasmodium* parasite motility is novel and will be very interesting to others in the community, however, several points have to be addressed; the work also needs to be discussed a bit more in the context of recently published data by other groups, who also reported in vitro motility assays and ATPase measurements for *Plasmodium* myosin A (e.g. Green et al JBC 2017).

We thank the reviewer for the positive assessment of our data.

The second paragraph of the Discussions (Lines 282-287) now discusses the work of Green et al., JBC 2017. 'The ATPase activity and unloaded in vitro motility speed of expressed PfMyoA were also recently measured by Green et al.³³, but with values for both that were over 10-fold lower than reported here. Although part of this discrepancy (~2-fold) can be attributed to different assay temperatures (our study 30°C, their study 23°C), and part to not knowing the state of phosphorylation of their expressed myosin (dephosphorylation would slow values 2-fold), it is unclear why their values are considerably lower than reported here.'

Main points:

1. Unfortunately, the crystal structures of the motor domains do not seem to include the lever arm with the light chains; the lever arm plus light chains seems to have been added in an estimated orientation, which makes this critical part of the structure difficult to assess;

line 60: “.. The amplitude of the subdomain movements and lever arm swing are similar to that of other myosins..”. Are there structural data on the lever arm and its swing between different nucleotide states available for PfMyoA and how do these structures compare to other myosins?

The first part of the lever arm is the converter subdomain which ends as a last helical structural element that usually continues as the main helix in the lever arm. Although we do not include data in this paper about the structure of the IQ motifs and its stabilization by the ELC and MTIP, we were able to model this lever arm in continuity with the last helix of the converter. So the estimation of the lever arm swing is from the converter swing which is similar in amplitude to that of other more classical myosins (such as Myo2, Myo5, Myo1...). We have now corrected the previous line 60 the following way:

‘In PfMyoA, the amplitude of the converter swing is similar to that observed for classical myosins’ (Line 193).

2. The authors estimate the amplitude of the myosin A power stroke from the modelled lever arm. It would be important to compare this estimate with the myosin A power stroke that has been measured in recent single molecule mechanical experiments using an optical trap (Green et al JBC 2017) and to discuss it in this context. Given the mechanical data, could the lever arm swing be smaller than that of other myosins for example?

The PfMyoA working stroke measured by Green et al was of 3 nm. *The MyoA “working stroke” was 3 nm, as measured from the shift in mean position of the observed event distribution* – This measurement was performed from myosin expressed using an *in vitro* transcription/translation system.

It is likely premature to estimate the full powerstroke amplitude before the structure of the lever arm is known. In previous measurement of the powerstroke for Myo10, Molloy et al reported a powerstroke in two steps of 15 nm and 2 nm (17 nm total) for Myo10HMM (amino acids 1–936) while the structural displacement predicted from structure would be 48 nm (Ropars et al. 2010) between the pre-powerstroke and the Rigor-like conformations. The working stroke measured in single molecule is thus not much smaller and not equivalent to the full powerstroke a myosin can adopt from the pre-powerstroke to the rigor-like structures.

3. The authors propose that phosphorylation of Ser 19 is serving as the switch to change the myosin motor mechanics between the sporozoite and the merozoite stage. Line 95: Ser19 phosphorylation occurs *in vivo* and tunes the PfMyoA properties. This is important for assessing the model (Fig. 4), and the authors should expand on this point; phosphorylation state of myosin A in sporozoites compared to merozoites? How does myosin A phosphorylation/

Currently there is no data about the effect of Ser19 *in vivo* on parasite motility. We have analysed the available proteomic data sets and added references to the appropriate papers to show that Ser19 phosphorylation has been found in each of the stages (4th paragraph in discussion), however these are not conclusive since the exact maturity/environmental context of the parasite will likely affect the phosphorylation states.

4. The difference in speed in the *in vitro* motility in the absence of phosphorylation and N-terminus is very interesting and should be compared with other modes of PfMyoA motor regulation, such as the light chains. The actin-activated ATPase experiments on myosin A should also be compared with the actin-activated ATPase data reported recently by other groups. What is the stoichiometry of the light chains MTIP and PfELC in the motility assays?

With regard to light chain regulation of PfMyoA, our prior work (Bookwalter et al., JBC 2017) showed that calcium binding had no effect on *in vitro* motility speed, so we have no evidence for regulation by the light chains i.e. calcium binding to the ELC. Given this result, we chose to not add anything in the Discussion as it would detract from the flow.

With regard to the work of Green et al., JBC 2017, the second paragraph of the Discussion now reads:

(Lines 282-287) 'The ATPase activity and unloaded *in vitro* motility of expressed PfMyoA was also recently measured by Green et al. (JBC, 2017), but with values for both that were over 10-fold lower than reported here. Although part of this discrepancy (~2-fold) can be attributed to different assay temperatures (our study 30°C, their study 23°C), and part to not knowing the state of phosphorylation of their expressed myosin (dephosphorylation would slow values 2-fold), it is unclear why their values are considerably lower than reported here.'

To ensure that both light chains are at full occupancy at the low concentrations of the motility assay, we added bacterially expressed PfELC and PfMTIP (each at 25 ug/ml) to the final assay buffer, as stated in

Methods.

Reviewers' Comments:

Reviewer #1:

Remarks to the Author:

The authors addressed all my previous comments satisfactorily. I congratulate them to their excellent work.

Stefan Raunser

Reviewer #3:

Remarks to the Author:

The authors have addressed my points in their response letter; furthermore, the manuscript has been thoroughly and satisfactorily revised and rewritten.

REVIEWERS' COMMENTS:

Reviewer #1 (Remarks to the Author):

The authors addressed all my previous comments satisfactorily. I congratulate them to their excellent work.

Stefan Raunser

Reviewer #3 (Remarks to the Author):

The authors have addressed my points in their response letter; furthermore, the manuscript has been thoroughly and satisfactorily revised and rewritten.

We thank the referees for the positive appreciation of our manuscript and also for useful discussions.